# D$^2$CSG: Unsupervised Learning of Compact CSG Trees with Dual Complements and Dropouts

Fenggen Yu[1]     Qimin Chen[1]     Maham Tanveer[1]
Ali Mahdavi Amiri[1]     Hao Zhang[1,2]
[1]Simon Fraser University     [2]Amazon

## Abstract

Abstract: We present D$^2$CSG, a neural model composed of two *dual* and *complementary* network branches, with *dropouts*, for unsupervised learning of *compact* constructive solid geometry (CSG) representations of 3D CAD shapes. Our network is trained to reconstruct a 3D shape by a fixed-order assembly of quadric primitives, with both branches producing a union of primitive intersections or inverses. A key difference between D$^2$CSG and all prior neural CSG models is its dedicated *residual branch* to assemble the potentially complex shape *complement*, which is subtracted from an overall shape modeled by the *cover* branch. With the shape complements, our network is provably general, while the weight dropout further improves compactness of the CSG tree by removing redundant primitives. We demonstrate both quantitatively and qualitatively that D$^2$CSG produces compact CSG reconstructions with superior quality and more natural primitives than all existing alternatives, especially over complex and high-genus CAD shapes.

## 1  Introduction

CAD shapes have played a central role in the advancement of geometric deep learning, with most neural models to date trained on datasets such as ModelNet [51], ShapeNet [2], and PartNet [33] for classification, reconstruction, and generation tasks. These shape collections all possess well-defined category or class labels, and more often than not, the effectiveness of the data-driven methods is tied to how well the class-specific shape features can be learned. Recently, the emergence of datasets of CAD *parts* and *assemblies* such as ABC [21] and Fusion360 [49] has fueled the need for learning shape representations that are *agnostic to class labels*, without any reliance on class priors. Case in point, the ABC dataset does not provide any category labels, while another challenge to the ensuing representation learning problem is the rich topological varieties exhibited by the CAD shapes.

*Constructive solid geometry* (CSG) is a classical CAD representation; it models a 3D shape as a recursive assembly of solid primitives, e.g., cuboids, cylinders, etc., through Boolean operations including union, intersection, and difference. Of particular note is the indispensable role the difference operation plays when modeling holes and high shape genera, which are common in CAD. Recently, there have been increased interests in 3D representation learning using CSG [9, 50, 41, 40, 19, 57, 6, 3], striving for generality, compactness, and reconstruction quality of the learned models.

In terms of primitive counts, a direct indication of compactness of the CSG trees, and reconstruction quality, CAPRI-Net [57] represents the state of the art. However, it is *not* a general neural model, e.g., it is unable to represent CAD shapes whose assembly necessitates *nested difference* operations (i.e., needing to subtract a part that requires primitive differences to build itself, e.g., see the CAD model in Fig. 2). Since both operands of the (single) difference operation in CAPRI-Net can only model intersections and unions, their network cannot produce a natural and compact CSG assembly for relatively complex CAD shapes with intricate concavities and topological details; see Fig. 1.

37th Conference on Neural Information Processing Systems (NeurIPS 2023).

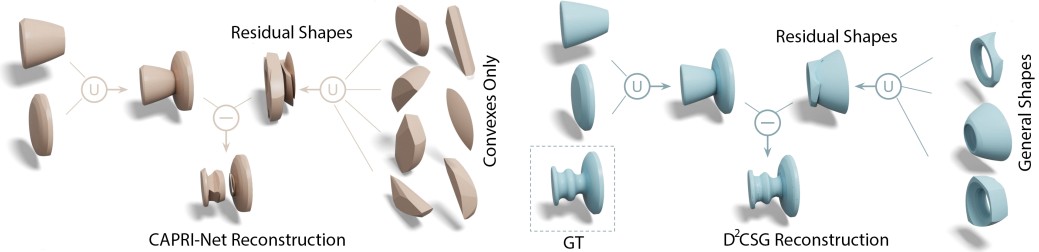

Figure 1: Comparing CSG trees and shape reconstructions by our network, D$^2$CSG, and CAPRI-Net, the current state of the art. To reproduce the ground truth (GT) shape, a natural CSG assembly necessitates a difference operation involving a complex residual shape, which D$^2$CSG can predict with compactness (only three intermediate, *general* shapes) and quality. CAPRI-Net can only build it using a union of convexes, requiring unnecessarily many primitives and leading to poor reconstruction.

In this paper, we present D$^2$CSG, a novel neural network composed of two *dual* and *complementary branches* for unsupervised learning of *compact* CSG tree representations of 3D CAD shapes. As shown in Fig. 2, our network follows a *fixed-order* CSG assembly, like most previous unsupervised CSG representation learning models [57, 6, 3]. However, one key difference to all of them is the *residual branch* that is dedicated to the assembly of the potentially complex *complement* or residual shape. In turn, the shape complement is subtracted from an overall shape that is learned by the *cover branch*. Architecturally, the two branches are identical, both constructing a union of intersections of quadric primitives *and* primitive inverses, but they are modeled by independent network parameters and operate on different primitive sets. With both operands of the final difference operation capable of learning general CAD shape assemblies, our network excels at compactly representing complex and high-genus CAD shapes with higher reconstruction quality, compared to the current state of the art, as shown in Fig. 1. To improve the compactness further, we implement a *dropout* strategy over network weights to remove redundant primitives, based on an *importance metric*.

Given the challenge of unsupervised learning of CSG trees amid significant structural diversity among CAD shapes, our network is not designed to learn a unified model over a shape collection. Rather, it *overfits* to a given 3D CAD shape by optimizing a compact CSG assembly of quadric surface primitives to approximate the shape. The learning problem is still challenging since the number, selection, and assembly of the primitives are unknown, inducing a complex search space.

In contrast to CAPRI-Net, our method is *provably general*, meaning that any CSG tree can be converted into an equivalent D$^2$CSG representation. Our dual branch network is fully differentiable and can be trained end-to-end with only the conventional occupancy loss for neural implicit models [4, 3, 57]. We demonstrate both quantitatively and qualitatively that our network, when trained on ABC [21] or ShapeNet [2], produces CSG reconstructions with superior quality, more natural trees, and better quality-compactness trade-off than all existing alternatives [3, 6, 57].

## 2   Related Work

In general, a 3D shape can be represented as a set of primitives or parts assembled together. Primitive fitting to point clouds has been extensively studied [29, 26, 24]. For shape abstraction, cuboid [44][54] and super quadrics [38] have been employed, while 3D Gaussians were adopted for template fitting [12]. From a single image, cuboids have also been used to estimate object parts and their relations using a convolutional-recursive auto-encoder [34]. More complex sub-shapes have been learned for shape assembly such as elementary 3D structures [8], implicit convex [7], and neural star components [20], as well as *parts* in the form of learnable parametric patches [41], moving or deformable primitives [30, 58, 55, 37], point clouds [28], or a part-aware latent space [10]. However, none of these techniques directly addresses reconstructing a CSG tree for a given 3D shape.

**Deep CAD Models.**   Synthesizing and editing CAD models is challenging due to their sharp features and non-trivial topologies. Learning-based shape programs have been designed to perform these tasks by providing easy-to-use editing tools [43, 11, 17, 1]. Boundary Representations (B-Reps) are common for CAD modeling and there have been attempts to *reverse engineer* such representations given an input mesh or point cloud [52]. For example, BRepNet [23], UV-Net [15], and SBGCN [16]

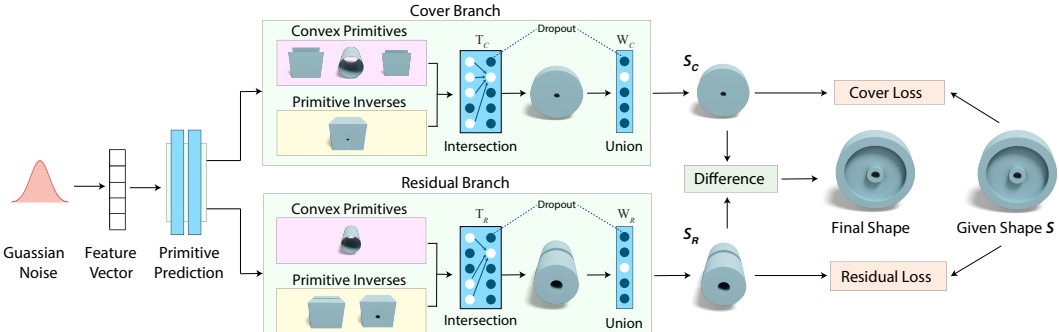

Figure 2: For a given 3D shape $S$ (shown at the right end), our network D$^2$CSG is trained to optimize both its network parameters and a feature code to reconstruct $S$ by optimizing an occupancy loss. The network parameters define a CSG assembly over a set of quadric primitives and primitives inverses. The assembly is built using two identical branches: a cover branch (top), producing shape $S_C$, and a residual branch (bottom), producing shape $S_R$. After applying intersections and a union to obtain the cover shape $S_C$ and residual shape $S_R$, by optimizing their respective occupancy losses, the recovered shape is obtained via a difference operation. A dropout is further applied to the parameters in the intersection and union layer for removing redundant primitives and intermediate shapes.

offer encoder-based networks to work with B-Reps and their topological information through message passing for CAD model analysis. NH-Rep [14] proposed several hand-crafted rules to infer CSG trees from B-Reps. However, the leaf nodes of their CSG trees are not parameterized primitives but implicit surfaces. Extrusion-based techniques [50, 53, 39] have also been used in deep-learning-based CAD shape generation, but extrusions do not provide general shape representations and the ensuing supervised learning methods [48, 50] are restricted to specific datasets.

**Learning CSG.**   Learning CSG representations, e.g., primitive assembly [47] and sketch analysis [35, 25], has become an emerging topic of geometric deep learning. While most approaches are supervised, e.g., SPFN [26], ParseNet [41], DeepCAD [50], ComplexGen [13], Lambourne et al. [22] and Point2Cyl [45], there have been several recent attempts at unsupervised CSG tree reconstruction, especially under the class-agnostic setting, resorting to neural implicit representations [4, 31, 36, 3]. InverseCSG [9] is another related method, which first detects a collection of primitives over the surface of an input shape using RANSAC, then performs a search to find CSG sequences to reconstruct the desired shape. However, RANSAC requires meticulous parameter tuning and the initially detected primitives can be noisy. Additionally, the space of CSG sequences is vast, leading to a lengthy search process. CSG-Net [40] employs a GRU-based decoder to generate CSG programs and resorts to RL to find the optimal CSG sequence for a target shape. However, the large search space for their beam search limits the method's ability to reconstruct complex target shapes. PLAD [18] follows CSG-Net and improves the program training with pseudo-labels and approximate distributions, but suffers from similar limitations as CSG-Net due to the use of similar program grammars and sequential network designs. UCSG-Net [19] learns to reconstruct CSG trees with arbitrary assembly orders. Such a task is difficult due to the order flexibility, but it can be made feasible by limiting the primitives to boxes and spheres only. More success in terms of reconstruction quality and compactness of the CSG trees has been obtained by learning *fixed-order* assemblies, including D$^2$CSG.

BSP-Net [3] learns plane primitives whose half-spaces are assembled via intersections to obtain convexes, followed by a union operation, to reconstruct concave shapes. CAPRI-Net [57] extends BSP-Net by adding quadric surface primitives and a difference operation after primitive unions. Similar to D$^2$CSG, CAPRI-Net [57] also tries to perform a final shape difference operation by predicting a set of primitives that undergo a fixed order of intersection, union, and difference operations to obtain the target shape. However, it constrains all primitives to be convex and its sequence is not *general* to support all shapes. To support generality, the set of primitives in D$^2$CSG includes both convex primitives and complimentary primitives (see Section 3.1). This enables the integration of difference operations during the initial phases of the CSG sequence. See the supplementary material for a formal proof of the generality of D$^2$CSG. CSG-Stump [6] also follows a fixed CSG assembly order while including an inverse layer to model shape complements. The complement operation helps attain generality of their CSG reconstructions, in theory, but the inverse

Table 1: Comparing various aspects of CAPRI-Net [57] and D²CSG.

|  | CAPRI [57] | D²CSG |
|---|---|---|
| Primitives | Convex quadrics | Convex quadrics and their inverses |
| # Difference operation | One | Multiple |
| Generalization | Certain shapes | All shapes |
| CSG layer parameters | ≈64K | ≈32K |
| Training time per shape | ≈3 mins | ≈5 mins |

layer is non-differentiable and the difference operations can only be applied to basic primitives, which can severely compromise the compactness and quality of the reconstruction. ExtrudeNet [39] follows the same CSG order as [6], but changes the modeling paradigm to extrusion to improve edibility. Later, SECAD [27] employs 2D implicit fields to produce more accurate 2D profiles for extrusion. The main issue with both methods is the lack of generality of the extrusion-based shapes.

**Overfit models.** Learning an overfit model to a single shape is not uncommon in applications such as compression [5], reconstruction [46], and level-of-details modeling [42] due to the high-quality results it can produce. The underlying geometry of a NeRF is essentially an overfit (i.e., fixed) to the shape/scene although the primary task of NeRF is novel view synthesis [56, 32]. Following a similar principle, to replicate fine geometric details, D²CSG constructs a CSG tree for a given object via optimizing a small neural network along with a randomly initialized feature code (Fig. 2). We follow this optimization/overfitting procedure as we did not find learning a prior on CAD shapes very useful and generalizable due to the structural and topological diversity in the CAD shapes.

## 3 Method

CSG involves performing operations on a set of *primitives* (leaves of a CSG tree), resulting in the creation of *intermediate shapes* that are then merged to form the *final shape*. In this work, sampled from a Gaussian distribution, the input to D²CSG is a feature vector that is optimized along with the network's parameters to reconstruct a given 3D shape $S$ and the output is a set of occupancy values that are optimized to fit $S$ (similar to Auto-decoder in DeepSDF [36]). We avoid encoding a shape to a latent code since it does not help to converge more accurately or faster (supplementary material). Our design is efficient as D²CSG uses a light network converging quickly to each shape.

As shown in Fig. 2, starting from a feature vector, we first pass it to the primitive prediction network to generate two matrices $\mathbf{P}_C$ and $\mathbf{P}_R$ that hold the primitives' parameters. Each matrix is used to calculate the point-to-primitive *approximated signed distance* (ASD) matrix $\mathbf{D}_C$ and $\mathbf{D}_R$ for query points sampled near the target shape. These two matrices are separately passed to the cover and residual branches, each having an intersection and a union layer. The two branches generate point occupancies $\mathbf{a}_C$ and $\mathbf{a}_R$ with respect to the cover and residual shapes $S_C$ and $S_R$, respectively. The final shape is obtained as the difference between $S_C$ and $S_R$. We adopt the multi-stage training of CAPRI-Net [57] to achieve accurate CSG reconstructions. Furthermore, since there is no explicit compactness constraint, many redundant primitives and intermediate shapes exist in the predicted CSG sequence. To this end, we introduce a novel dropout operation on the weights of each CSG layer to iteratively reduce redundancy and achieve higher level of compactness.

**Comparison to CAPRI-Net.** The most relevant work to our method is CAPRI-Net [57], which assembles a set of quadric primitives via a fixed order of intersection, union, and difference operations. We adopt the same equations as CAPRI-Net to define the convex quadric primitives and CSG operations. However, as highlighted in Table 1, our network, D²CSG, differs from CAPRI-Net in several aspects. First, D²CSG uses additional inverse primitives to support nested difference operations, which CAPRI-Net cannot handle. In general, despite of the fixed-order CSG operations, our network can reproduce the result of any CSG assembly, unlike CAPRI-Net, while using fewer network parameters. In addition, the new dropout design leads to more compact CSG assemblies.

### 3.1 Primitive Representation

The implicit quadric equation from CAPRI-Net [57] can cover frequently used *convex* primitives, e.g., planes, spheres, etc. However, its fixed CSG order (intersection, union, and difference) is not

general to combine the convexes to produce any shape; see proof in supplementary material. With D$^2$CSG, we introduce a more general primitive form to also cover *inverse* convex primitives, so that the difference operation can be utilized as the last, as well as the first, operation of the learned CSG sequence. This is one of the key differences between our method and CAPRI-Net [57], which only produces convex shapes from the intersection layer. Our primitive prediction network (an MLP) receives a code of size 256 and outputs two *separate* matrices $\mathbf{P}_C \in \mathbb{R}^{p \times 7}$ and $\mathbf{P}_R \in \mathbb{R}^{p \times 7}$ (for cover and residual branches), each contains parameters of $p$ primitives (see Fig. 2). We discuss the importance of separating $\mathbf{P}_C$ and $\mathbf{P}_R$ in the next section. Half of the primitives in $\mathbf{P}_C$ and $\mathbf{P}_R$ are convex represented by a quadric equation: $|a|x^2 + |b|y^2 + |c|z^2 + dx + ey + fz + g = 0$.

**Complementary Primitives.** In D$^2$CSG, the other half of the primitives in $\mathbf{P}_C$ and $\mathbf{P}_R$ are inverse convex primitives with the first three coefficients to be negative:

$$-|a|x^2 - |b|y^2 - |c|z^2 + dx + ey + fz + g = 0. \tag{1}$$

We avoid using a general quadric form without any non-negative constraints since it would produce complex shapes not frequently used in CAD design. For reconstruction, $n$ points near the shape's surface are sampled and their ASD to all primitives is calculated similarly to CAPRI-Net [57]. For each point $\mathbf{q}_j = (x_j, y_j, z_j)$, its ASD is captured in matrix $\mathbf{D} \in \mathbb{R}^{n \times p}$ as: $\mathbf{D}_C(j, :) = \mathbf{Q}(j, :)\mathbf{P}_C^T$ and $\mathbf{D}_R(j, :) = \mathbf{Q}(j, :)\mathbf{P}_R^T$, where $\mathbf{Q}(j, :) = (x_j^2, y_j^2, z_j^2, x_j, y_j, z_j, 1)$ is the $j$th row of $\mathbf{Q}$.

## 3.2 Dual CSG Branches

Previous CSG learning approaches employ a fixed CSG order and do not apply difference operation as the last operation as opposed to CAPRI-Net [57] and D$^2$CSG. Applying the difference as the last operation helps produce complex concavity for the reconstructed shape [57]. However, CAPRI-Net utilizes a shared primitive set for the two shapes that undergo the final subtraction operation. This sharing between the two operands has adverse effects on the final result. To backpropagate gradients through all the primitives, values in the selection matrix $\mathbf{T}$ in the intersection layer are float in the early training stage (Equation 2 and Table 2). Therefore many primitives simultaneously belong to cover and residual shapes even though their associated weights might be small in $\mathbf{T}$. Thus, the primitives utilized by the residual shape may impact the cover shape and result in the removal of details, or the opposite may also occur. Our dual branch in D$^2$CSG, however, allows the cover and residual shapes to utilize distinct primitives, enabling them to better fit their respective target shapes.

Now, we briefly explain our dual CSG branches using similar notations of BSP-Net [3]. We input the ASD matrix $\mathbf{D}_C$ into the cover branch and $\mathbf{D}_R$ into the residual branch, and output vector $\mathbf{a}_C$ and $\mathbf{a}_R$, indicating whether query points are inside/outside cover shape $S_C$ and residual shape $S_R$. Each branch contains an intersection layer and a union layer adopted from BSP-Net [3].

During training, the inputs to intersection layers are two ASD matrices $\mathbf{D}_R \in \mathbb{R}^{n \times p}$ and $\mathbf{D}_C \in \mathbb{R}^{n \times p}$. Primitives involved in forming intersected shapes are selected by two selection matrices $\mathbf{T}_C \in \mathbb{R}^{p \times c}$ and $\mathbf{T}_R \in \mathbb{R}^{p \times c}$, where $c$ is the number of intermediate shapes. The two selection matrices $\mathbf{T}_C$ and $\mathbf{T}_R$ are set as learnable weights in the intersection layer. We can obtain point-to-intermediate-shape indicator $\mathbf{Con} \in \mathbb{R}^{n \times c}$ by the intersection layer and only when $\mathbf{Con}(j, i) = 0$, query point $\mathbf{q}_j$ is inside the intermediate shape $i$ ($\mathbf{Con}_R$ is the same and only subscripts are $R$):

$$\mathbf{Con}_C = \text{relu}(\mathbf{D}_C)\mathbf{T}_C \quad \begin{cases} 0 & \text{in,} \\ > 0 & \text{out.} \end{cases} \tag{2}$$

Then all the shapes obtained by the intersection operation are combined by two union layers to find the cover shape $S_C$ and residual shape $S_R$. The inside/outside indicators of the combined shape are stored in vector $\mathbf{a}_R$ and $\mathbf{a}_C \in \mathbb{R}^{n \times 1}$ indicating whether a point is in/outside of the cover and residual shapes. We compute $\mathbf{a}_C$ and $\mathbf{a}_R$ in a multi-stage fashion ($\mathbf{a}^+$ and $\mathbf{a}^*$ for early and final stages) as [57]. Specifically, $\mathbf{a}_C^*$ and $\mathbf{a}_R^*$ are obtained by finding min of each row of $\mathbf{Con}_C$ and $\mathbf{Con}_R$:

$$\mathbf{a}_C^*(j) = \min_{1 \leq i \leq c}(\mathbf{Con}_C(j, i)) \quad \begin{cases} 0 & \text{in,} \\ > 0 & \text{out.} \end{cases} \tag{3}$$

Since gradients can only be backpropagated to the minimum value in the min operation, we additionally introduce $\mathbf{a}_C^+$ at the early training stage to facilitate learning by the following equation:

Table 2: Settings for multi-stage training.

| Stage | Intersection ($\mathbf{T}$) | Union ($\mathbf{W}$) | Difference Op ($\mathbf{a}$) | Dropout | Loss |
|---|---|---|---|---|---|
| 0 | float | float | $\mathbf{a}^+$ | - | $L_{rec}^+ + L_{\mathbf{T}} + L_{\mathbf{W}}$ |
| 1 | float | - | $\mathbf{a}^*$ | - | $L_{rec}^* + L_{\mathbf{T}}$ |
| 2 | binary | binary | $\mathbf{a}^*$ | ✓ | $L_{rec}^*$ |

$$\mathbf{a}_C^+(j) = \mathscr{C}\left(\sum_{1 \le i \le c} \mathbf{W}_C(i)\mathscr{C}(1 - \mathbf{Con}_C(j,i))\right) \begin{cases} 1 & \approx \text{in,} \\ < 1 & \approx \text{out.} \end{cases} \tag{4}$$

$\mathbf{W}_C \in \mathbb{R}^c$ is a learned weighting vector and $\mathscr{C}$ is a clip operation to $[0, 1]$. $\mathbf{a}_R^+$ is defined similarly.

### 3.3 Loss Functions and Dropout in Training

Our training process is similar to that of CAPRI-Net [57] in terms of staging. The main motivation behind this is to ensure that the values in the CSG layer weights gradually change from float values to binary values, while facilitating better gradient propagation for learning the CSG layer weights in the early stage. While the early stages (0 - 1) of our training and loss functions are the same as CAPRI-Net, stage 2, which is a crucial part of our training due to the injection of a novel dropout, is different. In Table 2, we provide an overview of each stage of training and loss functions, with more details presented below and also in the supplementary material.

**Stage 0.** At stage 0, $\mathbf{T}$ in the intersection layer and $\mathbf{W}$ in the union layer are float. The loss function is $L^+ = L_{rec}^+ + L_{\mathbf{T}} + L_{\mathbf{W}}$, where $L_{\mathbf{T}}$ and $L_{\mathbf{W}}$ serve to limit the entries of $\mathbf{T}$ to be between 0 and 1, and the entries of $\mathbf{W}$ to be almost 1. This setup would help the union layer avoid using the min operation in Equation 8 at stage 0 so that all the intermediate shapes could have gradients. Specifically,

$$L_{\mathbf{T}} = \sum_{t \in T} \max(-t, 0) + \max(t - 1, 0), L_{\mathbf{W}} = \sum_i |\mathbf{W}_i - 1|. \tag{5}$$

It is worth noting that we have separate network weights for the dual CSG branches: $\mathbf{T} = [\mathbf{T}_C, \mathbf{T}_R]$ and $\mathbf{W} = [\mathbf{W}_C, \mathbf{W}_R]$, and they are trained separately. Our reconstruction loss $L_{rec}^+$ is a combination of the cover loss $L_C^+$ and the residual loss $L_R^+$, which are defined below.

$$L_C^+ = \frac{1}{n} \sum_{j=1}^n \mathbf{M}_C(j) * (\mathbf{g}(j) - \mathbf{a}_C^+(j))^2, \tag{6}$$

$$L_R^+ = \frac{1}{n} \sum_{j=1}^n \mathbf{M}_R(j) * ((1 - \mathbf{g}(j)) - \mathbf{a}_R^+(j))^2, \tag{7}$$

where $\mathbf{g}$ denotes the ground truth (GT) point occupancy values. $\mathbf{M}_C(j) = \max(\mathbf{g}(j), \mathbb{1}(\mathbf{a}_R^+(j) < \beta))$ and $\mathbf{M}_R(j) = \max(\mathbf{g}(j), \mathbb{1}(\mathbf{a}_C^+(j) > \beta))$, where $\beta$ is set to be 0.5 to signify the shape surface at stage 0. The function $\mathbb{1}$ maps Boolean values to float, while $\mathbf{M}_C$ and $\mathbf{M}_R$ adjust the losses on each branch with respect to the value of the other branch. In the supplementary material, we provide 2D visualization examples to help comprehend these various notations.

**Stage 1.** Here, the vector $\mathbf{a}^+$ is substituted with $\mathbf{a}^*$ to eliminate the use of $\mathbf{W}$ from stage 0. This adopts min as the union operation for the cover and residual shapes and restricts the back-propagation of gradients solely to the shapes that underwent a union operation. Note that $\mathbf{W}$ is not updated at this stage. The loss function is $L^* = L_{rec}^* + L_T$, where $L_T$ is the same as in stage 0, and $L_{rec}^*$ is the reconstruction loss: $L_{rec}^* = L_C^* + L_R^*$. The cover loss $L_C^*$ and residual loss $L_R^*$ are defined as:

$$L_C^* = \frac{1}{n} \sum_{j=1}^n \mathbf{M}_C(j) * [(1 - \mathbf{g}(j)) * (1 - \mathbf{a}_C^*(j)) + w_C * \mathbf{g}(j) * \mathbf{a}_C^*(j)], \tag{8}$$

$$L_R^* = \frac{1}{n} \sum_{j=1}^n \mathbf{M}_R(j) * [\mathbf{g}(j) * (1 - \mathbf{a}_R^*(j)) + w_R * (1 - \mathbf{g}(j)) * \mathbf{a}_R^*(j)]. \tag{9}$$

Here, $\mathbf{M}_C(j) = \max(\mathbf{g}(j), \mathbb{1}(\mathbf{a}_R^*(j) > \gamma))$, $\mathbf{M}_R(j) = \max(\mathbf{g}(j), \mathbb{1}(\mathbf{a}_C^*(j) < \gamma))$, $\mathbf{g}$ is the GT point occupancy value operating as a mask, and $\gamma = 0.01$ is close to the inside value 0. We choose to use the $L_1$ loss function in stage 1 so that the different weights can be applied to encourage utilization of the difference operation. In addition, we set $w_C = 10$ and $w_R = 2.5$, as in CAPRI-Net [57], to encourage the cover and residual shapes to fulfill their roles in the reconstruction.

**Stage 2.** In stage 2, entries $t$ in $T$ are quantized into binary values by a threshold, $t_{Binary} = (t > \eta)?1 : 0$, where $\eta = 0.01$. This way, the network will perform interpretable intersection operations on primitives. Due to the non-differentiable nature of the $t_{Binary}$ values, the loss term $L_T$ is no longer utilized, resulting in the reduction of the loss function to only $L_{rec}^*$ (see Table 2).

**Dropout in Stage 2.** Without an explicit compactness constraint or loss, redundancies in our CSG reconstruction are inevitable. To alleviate this, we make two crucial observations: first, the CSG sequence learned at stage 2 is entirely interpretable, allowing us to track the primitives and CSG operations involved in creating the final shape; second, altering the primitives or CSG operations will consequently modify the final shape's implicit field. Therefore, we can identify *less significant* primitives and intermediate shapes if their removal does not substantially impact the final result. Thus, we introduce the importance metric $\Delta \mathbf{S}$ to measure the impact of removing primitives or intermediate shapes on the outcome. We first define the implicit field $\mathbf{s}^*$ of the final shape as follows:

$$\mathbf{s}^*(j) = \max(\mathbf{a}_C^*(j), \alpha - \mathbf{a}_R^*(j)) \begin{cases} 0 & \text{in,} \\ > 0 & \text{out,} \end{cases} \tag{10}$$

where $\mathbf{s}^*(j)$ signifies whether a query point $\mathbf{q}_j$ is inside or outside the reconstructed shape. It is important to note that $\alpha$ should be small and positive ($\approx 0.2$), as $\mathbf{s}^*$ approaches 0 when points are within the shape. Then the importance metric $\Delta \mathbf{S}$ is defined as:

$$\Delta \mathbf{S} = \sum_{j=1}^{n} \Delta \mathbb{1}(\mathbf{s}^*(j) < \frac{\alpha}{2}). \tag{11}$$

Here, $\mathbf{s}^*(j) < \frac{\alpha}{2}$ quantizes implicit field values $\mathbf{s}^*(j)$ from float to Boolean, where inside values are 1 and outside values are 0. Function $\mathbb{1}$ maps Boolean values to integers for the sum operation, $\Delta$ captures the difference in values in case of eliminating primitives or intermediate shapes.

Note that union layer parameters $\mathbf{W}$ are set close to 1 by $L_{\mathbf{W}}$ in stage 0. During stage 2, at every 4,000 iterations, if removing the intermediate shape $i$ from the CSG sequence makes $\Delta \mathbf{S}$ smaller than threshold $\sigma$, we would discard intermediate shape $i$ by making $\mathbf{W}_i = 0$. Consequently, Equation (6) will be modified to incorporate the updated weights in the union layer:

$$\mathbf{a}_C^*(j) = \min_{1 \leq i \leq c} (\mathbf{Con}_C(j, i) + (1 - \mathbf{W}_i) * \theta) \begin{cases} 0 & \text{in,} \\ > 0 & \text{out.} \end{cases} \tag{12}$$

This way, the removed shape $i$ will not affect $\mathbf{a}^*$ as long as $\theta$ is large enough. In our experiments, we discover that $\theta$ just needs to be larger than $\alpha$ so that the removed shape $i$ will not affect $\mathbf{s}^*$. Thus we set $\theta$ as 100 in all experiments. Furthermore, we apply dropout to parameters in the intersection layer by setting $T_{k,:} = 0$ if $\Delta \mathbf{S}$ falls below $\sigma$ after removing primitive $k$ from the CSG sequence. We set $\sigma$ as 3 in experiments and examine the impact of $\sigma$ in the supplementary material. After dropouts, the network parameters will be tuned. We iteratively perform this process until the maximum number of iterations has been reached or when no primitives or intermediate shapes are dropped.

## 4   Results

In our experiments to be presented in this section, we set the number of maximum primitives as $p = 512$ and the number of maximum intersections as $c = 32$ for each branch to support complex shapes. The size of our latent code is 256 and a two-layer MLP is used to predict the parameters of the primitives from the input feature code. We train $D^2CSG$ per shape by optimizing the latent code, primitive prediction network, intersection layer, and union layer. To evaluate $D^2CSG$ against prior methods, which all require an additional time-consuming optimization at test time to achieve satisfactory results (e.g., 30 min per shape for CSG-Stump), we have randomly selected a moderately sized subset of shapes as test set for evaluation: 500 shapes from ABC, and 50 from each of the 13 categories of ShapeNet (650 shapes in total). In addition, we ensured that 80% of the selected shapes from ABC have genus larger than two with more than 10K vertices to include complex structures. All experiments were performed using an Nvidia GeForce RTX 2080 Ti GPU.

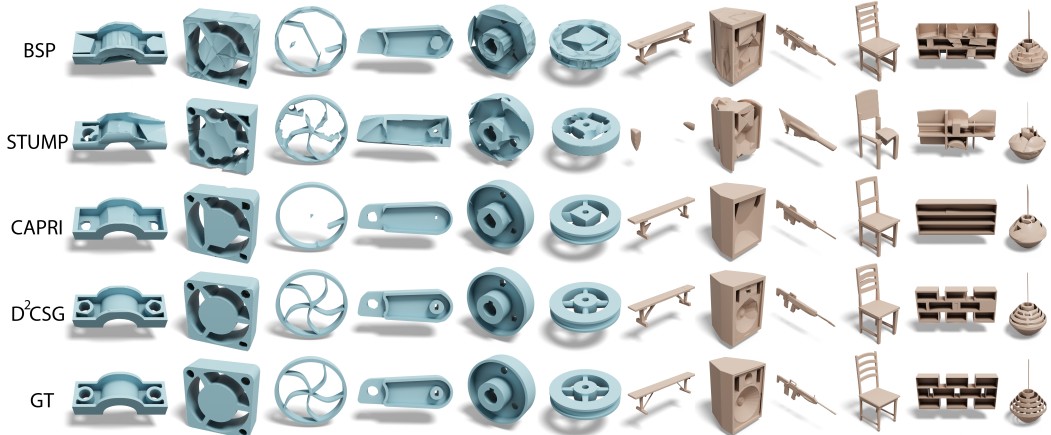

Figure 3: Comparing CSG representation learning and reconstruction from 3D meshes in ABC dataset (columns 1-6) and ShapeNet (columns 7-12). Results are best viewed when zooming in.

## 4.1 Mesh to CSG Representation

Given a 3D shape, the task is to learn an accurate and compact CSG representation for it. To do so, we first sample 24,576 points around the shape's surface (i.e. with a distance up to $1/64$) and 4,096 random points in 3D space. All 28,672 points are then scaled into the range $[-0.5, 0.5]$, these points along with their occupancy values are used to optimize the network.

We compare $D^2$CSG with InverseCSG [9] (I-CSG), BSP-Net [3], CSG-Stump [6] and CAPRI-Net [57], which output structured parametric primitives. For a fair comparison, we optimize all of these networks with the same number of iterations. BSP-Net, CSG-Stump, and CAPRI-Net are pre-trained on the training set provided in CAPRI-Net before optimization to achieve better initialization. Note that CSG-Stump uses different network settings for shapes from ABC (with shape differences) and ShapeNet (without shape difference); we therefore follow the same settings in our comparisons. For InverseCSG, we adopt the RANSAC parameters used for most of shapes in its CAD shape dataset. For each shape reconstruction, BSP-Net takes about 15 min, I-CSG about 17 min, CSG-Stump about 30 min, and CAPRI-Net about 3 min to converge. In contrast, the training process for $D^2$CSG runs 12,000 iterations for each stage, taking about 5 minutes per shape reconstruction.

**Evaluation Metrics.** Quantitative metrics for shape reconstruction are symmetric Chamfer Distance (CD), Normal Consistency (NC), Edge Chamfer Distance [3] (ECD). For ECD, we set the threshold for normal cross products to 0.1 for extracting points close to edges. CD and ECD are computed on 8K sample points on the surface and multiplied by 1,000. In addition, we compare the number of primitives #P to evaluate the compactness of shapes since all CSG-based modeling methods predict some primitives that are combined with intersection operations.

**Evaluation and Comparison.** We provide visual comparisons on representative examples from the ABC dataset and the ShapeNet dataset in Fig. 3. Since InverseCSG only accepts high-quality CAD meshes as input for its primitive detection algorithm and fails on ShapeNet, we only compare it on the ABC dataset, more visual results can be found in the supplementary material. Our method consistently reconstructs more accurate shapes with geometric details and concavities. The RANSAC based primitive detection algorithm in InverseCSG can easily produce noisy and inaccurate primitives, resulting in many degenerated planes after CSG modeling. BSP-Net simply assembles convex shapes to fit the target shape and obtain less compact results without a difference operation. CSG-Stump tends to use considerably more difference operations to reconstruct shapes. This also causes the shapes' surface to be carved by many redundant primitives (i.e., lack of compactness). In addition, since CSG-Stump does not support difference operations between complex shapes, it fails to reconstruct small holes or intricate concavities. As the intermediate shapes subtracted in CAPRI-Net's fixed CSG sequence are only unions of convex shapes, it fails to reproduce target shapes with complex concavities. $D^2$CSG achieves the best reconstruction quality and compactness in all metrics on ABC dataset and ShapeNet compared to other methods, except for NC on ShapeNet where $D^2$CSG underperformed ever so slightly ($\Delta = 0.003$); see Table 3.

Table 3: Comparing CSG rep learning from 3D meshes in ABC and ShapeNet.

| Methods | ABC | | | | | ShapeNet | | | |
|---|---|---|---|---|---|---|---|---|---|
| | I-CSG | BSP | STUMP | CAPRI | Ours | BSP | STUMP | CAPRI | Ours |
| CD ↓ | 0.576 | 0.115 | 0.383 | 0.177 | **0.069** | 0.164 | 2.214 | 0.124 | **0.119** |
| NC ↑ | 0.877 | 0.921 | 0.850 | 0.903 | **0.928** | 0.882 | 0.794 | **0.890** | 0.887 |
| ECD ↓ | 6.330 | 4.047 | 8.881 | 3.990 | **3.091** | 3.899 | 6.101 | 2.035 | **1.722** |
| #P ↓ | 43.62 | 359.38 | 83.42 | 66.26 | **28.62** | 694.21 | 228.58 | 50.94 | **43.78** |

Table 4: Ablation study on key components of $D^2CSG$: complementary primitives (CP), dual branches (DB), and dropout (DO), on three quality metrics and three compactness metrics, the number of CSG primitives (#P), intermediate shapes (#IS), and surface segments (#Seg) resulting from a decomposition induced by the CSG tree. Winner in boldface and second place in blue.

| Row ID | CP | DB | DO | CD ↓ | NC ↑ | ECD ↓ | #P ↓ | #IS ↓ | #Seg ↓ |
|---|---|---|---|---|---|---|---|---|---|
| 1 | - | - | - | 0.183 | 0.907 | 3.92 | 77 | 9.2 | 93 |
| 2 | ✓ | - | - | 0.073 | 0.935 | 3.12 | 38 | 5.8 | 55 |
| 3 | ✓ | - | ✓ | 0.088 | 0.926 | 3.48 | **27** | **5.3** | **40** |
| 4 | ✓ | ✓ | - | **0.069** | **0.936** | **2.98** | 53 | 6.8 | 57 |
| 5 | ✓ | ✓ | ✓ | **0.069** | 0.928 | 3.09 | 29 | 5.7 | 42 |

Table 5: Comparing CSG rep learning from 3D point cloud in ABC and ShapeNet.

| Methods | ABC | | | | ShapeNet | | | |
|---|---|---|---|---|---|---|---|---|
| | BSP | STUMP | CAPRI | Ours | BSP | STUMP | CAPRI | Ours |
| CD ↓ | 0.133 | 0.695 | 0.225 | **0.085** | 0.268 | 1.177 | 0.242 | **0.224** |
| NC ↑ | 0.919 | 0.841 | 0.894 | **0.924** | **0.889** | 0.840 | 0.888 | 0.886 |
| ECD ↓ | 3.899 | 7.303 | 3.308 | **3.029** | 1.854 | 3.482 | 1.971 | **1.815** |
| #P ↓ | 360.87 | 67.436 | 68.57 | **35.28** | 553.76 | 211.36 | 55.21 | **51.26** |

**Ablation Study.**   We conducted an ablation study (see Table 4) to assess the efficacy of, and tradeoff between, the three key features of $D^2CSG$: complementary primitives, dual branch design, and dropout. As we already have three metrics for reconstruction quality, we complement #P, the only compactness measure, with two more: the number of intermediate shapes and the number of surface segments as a result of shape decomposition by the CSG tree; see more details on their definitions in the supplementary material. We observe that incorporating complementary primitives (row 2) enables better generalization to shapes with complex structures and higher accuracy than the CAPRI-Net [57] baseline (row 1). The dual branch design further enhances reconstruction accuracy (row 4 vs. 2), as primitives from the residual shape do not erase details of the cover shape, and each branch can better fit its target shape without impacting the other branch. However, the dual branch design did compromise compactness. Replacing dual branching by dropout in the final stage (row 3 vs. 4) decreases the primitive count by about 50% and improves on the other two compactness measures, while marginally compromising accuracy. The best trade-off is evidently achieved by combining all three components, as shown in row 5, where adding the dual branches has rather minimal effects on all three compactness metrics with dropouts. In the supplementary material, we provide additional ablation studies. We also examine the impact of employing basic primitives, as well as modifying the maximum primitive count and the number of sampled points.

## 4.2   Applications

**Point Clouds to CSG.**   We reconstruct CAD shapes from point clouds, each containing 8,192 points. For each input point, we sample 8 points along its normal with perturbations sampled from Gaussian distribution ($\mu = 0, \sigma = 1/64$). If this point is at the opposite direction of normal vectors, the occupancy value is 1, otherwise it is 0. This way, we sample $65, 536$ points to fit the network to each shape and other settings the same as mesh-to-CSG experiment. Quantitative comparisons in Table 5 and visual comparisons in Fig. 4 show that our network outperforms BSP-Net, CSG-Stump, and CAPRI-Net in different reconstruction similarity and compactness metrics on ABC dataset and ShapeNet. More results can be found in the supplementary material.

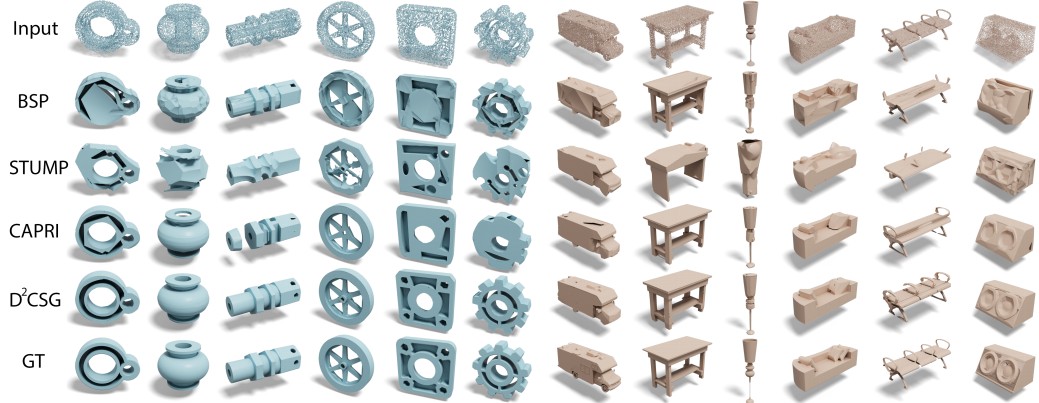

Figure 4: CSG reconstruction from 3D point cloud in ABC (Col. 1-6) and ShapeNet (Col. 7-12).

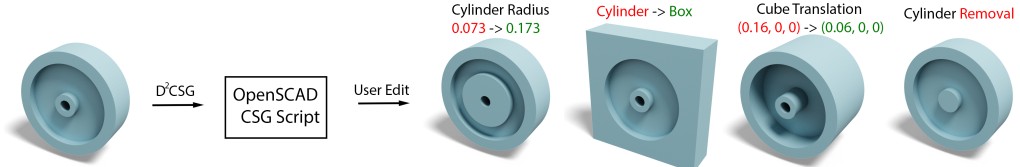

Figure 5: D$^2$CSG learns OpenSCAD scripts for a given shape and supports editability.

**Shape Editing.** Once we have acquired the primitives and CSG assembly operations, we can edit the shape by altering primitive parameters, adjusting the CSG sequence, or transforming intermediate shapes. To facilitate shape editing in popular CAD software, we further convert quadric primitives into basic primitives (e.g., cubes, spheres, cylinder, etc.) by identifying the best-fitting basic primitive for each predicted quadric primitive by our method. Subsequently, we can export the primitive parameters and CSG sequence into an OpenSCAD script. Fig. 5 shows the shape editing results.

## 5 Discussion, Limitation, and Future Work

We present D$^2$CSG, a simple yet effective idea, for unsupervised learning of general and compact CSG tree representations of 3D CAD objects. Extensive experiments on the ABC and ShapeNet datasets demonstrate that our network outperforms state-of-the-art methods both in reconstruction quality and compactness. We also have ample visual evidence that the CSG trees obtained by our method tend to be more compact than those produced by prior approaches.

Our network does not generalize over a shape collection; it "overfits" to a single input shape and is in essence an optimization to find a CSG assembly. While arguably limiting, this is not entirely unjustified since some of the models in the ABC dataset may not possess sufficient generalizability in their primitive assemblies. Another limitation is that our current quadric primitive representation cannot fully support complex surfaces, such as tori or NURBS. It would be an interesting future direction to change quadric primitives to NURBS or other easily editable primitives, such as those from extrusions. In addition, incorporating interpretable CSG operations into the network tends to cause gradient back-propagation issues and limits the reconstruction accuracy of small details such as decorative curves on chair legs. We would also like to extend our method to structured CAD shape reconstruction from images and free-form sketches. Another interesting direction for future work is to scale the primitive assembly optimization from CAD parts to indoor scenes.

## 6 Acknowledgement

We thank all the anonymous reviewers for their valuable comments, and KOKONI3D (https://www.kokoni3d.com/) for the helpful discussions. This work was supported in part by an NSERC Discovery Grant (No.611370) and industrial gift funds from Adobe and Autodesk.

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
