# $\text{D}^2\text{CSG}$: Unsupervised Learning of Compact CSG Trees with Dual Complements and Dropouts

**Fenggen Yu**[1]    **Qimin Chen**[1]    **Maham Tanveer**[1]
**Ali Mahdavi Amiri**[1]    **Hao Zhang**[1]
[1]Simon Fraser University

This supplementary document provides 1) two propositions and their proofs regarding the generality of $\text{D}^2\text{CSG}$ and CAPRI-Net in Section A; 2) additional details related to the notations and loss functions in Section B; 3) details about CAD mesh construction process and more shape editing results in Section C; 4) additional experiments regarding to ablation studies, pre-training, primitive choices and hyper-parameters in Section D.

## A  Generality Proofs

Here, we provide two propositions and their proofs regarding the generality of $\text{D}^2\text{CSG}$ and CAPRI-Net mentioned in Section 3 of the main paper.

**Proposition 1.** *The fixed order introduced by CAPRI-Net cannot support all possible combinations of CSG operations applied to convex shapes.*

*Proof.* To prove Proposition 1, we provide an example that CAPRI-Net's sequence fails to support. In Fig. 1, the union of two rings is produced using four given circular convex shapes $C_i$ as $S = (C_1 - C_2) \cup (C_3 - C_4)$. We first assume that we can convert it to the fixed operation sequence of CAPRI-Net where the last operation is a difference: $S = S_l - S_r$. There will be three cases:

Case 1: $S_l = C_1 - (C_2 - C_3)$, $S_r = C_4$ (Fig. 1 (b)).

Case 2: $S_l = C_1$, $S_r = (C_2 - C_3) \cup C_4$ (Fig . 1 (c)).

Case 3: $S_l = C_1 - C_4$, $S_r = C_2 - C_3$ (Fig. 1 (d)).

In all these cases, a difference operation is required either in $S_l$ or $S_r$. However, $S_l$ and $S_r$ can only be produced by intersection and union operations in CAPRI-Net's sequence. Note that CAPRI-Net does not support inverse convex shapes. Since we have found a contradiction, $S$ is not reproducible by CAPRI-Net's operation sequence and therefore the proposition is proven. $\square$

Before presenting the second proposition about the generality of $\text{D}^2\text{CSG}$, we provide three rules that are later needed for this proof (see Fig. 2 for an illustration).
**Rule 1:** $(A \cup C) - (B - C) = (A - B) \cup C$

*Proof.* $(A \cup C) - (B - C) = (A \cup C) - (B \cap \overline{C}) = (A \cup C) \cap \overline{(B \cap \overline{C})} = (A \cup C) \cap (\overline{B} \cup C) = ((A \cup C) \cap \overline{B}) \cup ((A \cup C) \cap C) = ((A \cup C) \cap \overline{B}) \cup C = ((A \cap \overline{B}) \cup (C \cap \overline{B})) \cup C = (A \cap \overline{B}) \cup C = (A - B) \cup C$

$\square$

**Rule 2:** $A - (B \cup C) = A - B - C$

*Proof.* $A - (B \cup C) = A \cap \overline{(B \cup C)} = A \cap (\overline{B} \cap \overline{C}) = A \cap \overline{B} \cap \overline{C} = A - B - C$

37th Conference on Neural Information Processing Systems (NeurIPS 2023).

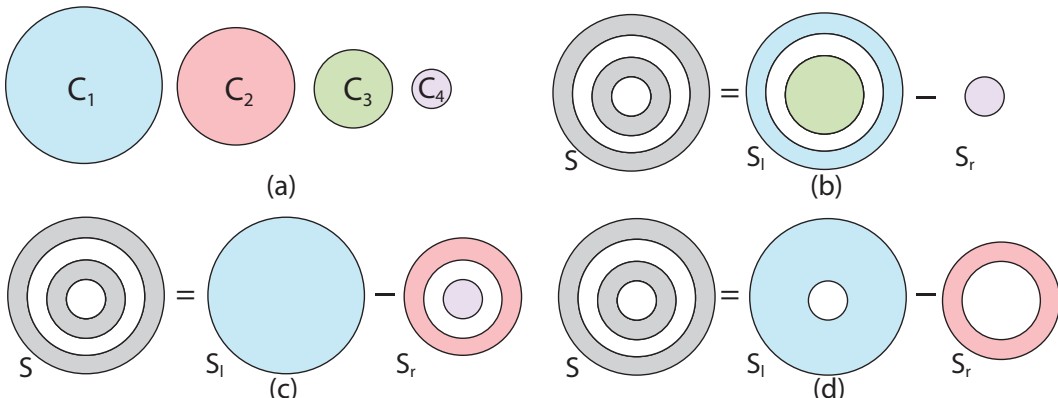

Figure 1: Given four convex shapes in (a), there are three ways to produce the double ring (Left) illustrated in (b), (c) and (d).

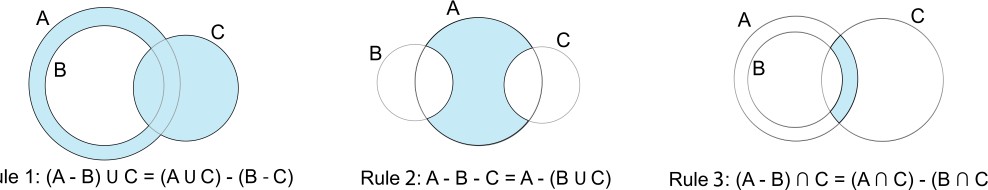

Rule 1: (A - B) ∪ C = (A ∪ C) - (B - C)   Rule 2: A - B - C = A - (B ∪ C)   Rule 3: (A - B) ∩ C = (A ∩ C) - (B ∩ C)

Figure 2: The three CSG rules we used in our proof.

$\square$

**Rule 3:** $(A \cap C) - (B \cap C) = (A - B) \cap C$

*Proof.* $(A \cap C) - (B \cap C) = (A \cap C) \cap \overline{(B \cap C)} = (A \cap C) \cap (\overline{B} \cup \overline{C}) = (A \cap C \cap \overline{B}) \cup (A \cap C \cap \overline{C}) = A \cap C \cap \overline{B} = A \cap \overline{B} \cap C = (A - B) \cap C$ $\square$

**Proposition 2.** *The operation sequence in $D^2$CSG is able to support any CSG sequence.*

*Proof.* Knowing that any CSG sequence can be represented by a binary CSG tree, we would use induction to prove that any binary CSG tree can be converted to our $D^2$CSG order, where the final operation is a difference between the cover shape $S_C$ and residual shape $S_R$.

**Base Cases:** We first prove that any single CSG operation based on two convex primitives $P_1$ and $P_2$ can be converted to our $D^2$CSG representation:

$P_1 \cap P_2 = (P_1 \cap P_2) - \emptyset \rightarrow S_C = (P_1 \cap P_2), S_R = \emptyset$
$P_1 \cup P_2 = (P_1 \cup P_2) - \emptyset \rightarrow S_C = (P_1 \cup P_2), S_R = \emptyset$
$P_1 - P_2 = P_1 - P_2 \rightarrow S_C = P_1, S_R = P_2$

As a result, it is evident that any single CSG operation based on two convex primitives can be converted to our $D^2$CSG representation, where the final operation is a difference operation between the cover shape $S_C$ and the residual shape $S_R$.

**Inductive Step:** Assume any CSG tree with $n$ CSG operations can be converted to our $D^2$CSG representation, where the final operation is a difference operation between the cover shape and the residual shape. Having a CSG tree $B$ with $n + 1$ CSG operations, we need to prove that $B$ can be converted to our $D^2$CSG representation.

We observe that tree $B$ can be split into two sub-trees at the root node: $B = B_C \odot B_R$, where $\odot$ is the CSG operation at the root node. Obviously, each of two subtrees $B_C$ and $B_R$ contains maximum $n$ CSG operations and can be represented by our $D^2$CSG representation: $B_C = \alpha_C - \alpha_R$ and $B_R = \beta_C - \beta_R$, where cover shapes $\alpha_C$ and $\beta_C$ are constructed only by intersection and

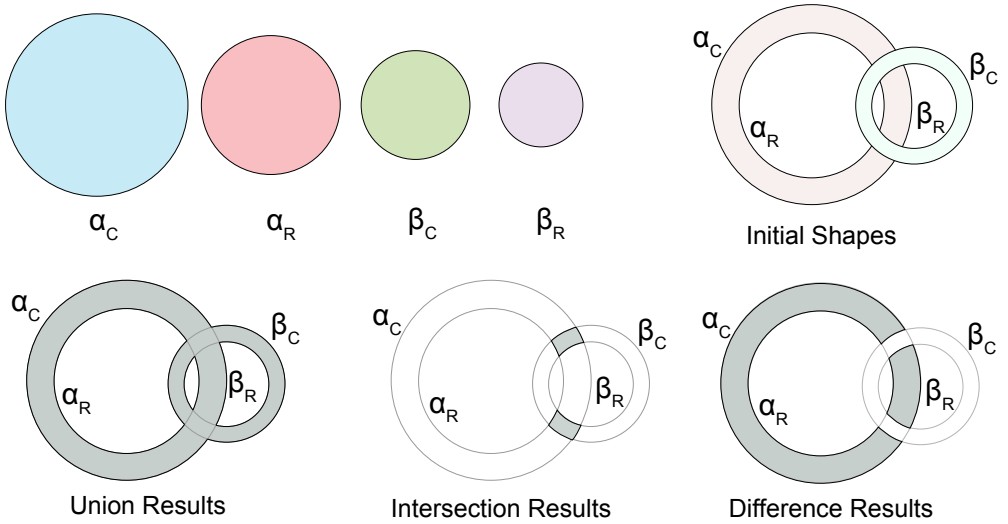

Figure 3: 2D examples for different CSG operations.

union operations, residual shapes $\alpha_R$ and $\beta_R$ are constructed only by inverse, intersection and union operations.

Here, we discuss all possible cases for operation $\odot$ between $B_C$ and $B_R$, see examples for all three cases in Fig. 3. We prove that all cases can be converted to our D$^2$CSG representation.

**Union case:** $B$ is a union of $B_C$ and $B_R$:

$$B = B_C \cup B_R = (\alpha_C - \alpha_R) \cup (\beta_C - \beta_R) \overset{R1}{=}$$
$$(\alpha_C \cup (\beta_C - \beta_R)) - (\alpha_R - (\beta_C - \beta_R))$$

Here, we have $S_C = (\alpha_C \cup (\beta_C - \beta_R))$ and $S_R = (\alpha_R - (\beta_C - \beta_R))$. Therefore, union case can be written in the form of D$^2$CSG representation.

**Intersection case:** $B$ is an intersection of $B_C$ and $B_R$:

$$B = B_C \cap B_R = (\alpha_C - \alpha_R) \cap (\beta_C - \beta_R) \overset{R3}{=}$$
$$(\alpha_C \cap (\beta_C - \beta_R)) - (\alpha_R \cap (\beta_C - \beta_R))$$

Here, $S_C = (\alpha_C \cap (\beta_C - \beta_R))$ and $S_R = (\alpha_R \cap (\beta_C - \beta_R))$.

**Difference case:** $B$ is a difference between $B_C$ and $B_R$:

$$B = B_C - B_R = (\alpha_C - \alpha_R) - (\beta_C - \beta_R) \overset{R2}{=}$$
$$(\alpha_C) - (\alpha_R \cup (\beta_C - \beta_R)).$$

Here, $S_C = \alpha_C$ and $S_R = \alpha_R \cup (\beta_C - \beta_R)$.

In all the above cases, each CSG operation between $B_C$ and $B_R$ can be converted to our D$^2$CSG representation, where the final operation is a difference operation between the cover shape $S_C$ and residual shape $S_R$. In addition, the cover shape $S_C$ and residual shape $S_R$ in the above three cases can be represented by three fixed order CSG layers(Inverse, Intersection and Union) from each branch according to the proof in CSG-Stump [1].

All the above derivations show that CSG tree $B$ with $n + 1$ CSG operations can also be converted to our D$^2$CSG representation, where the final operation is a difference operation between the cover shape $S_C$ and residual shape $S_R$. □

## B  2D Examples for Notations and Loss Functions

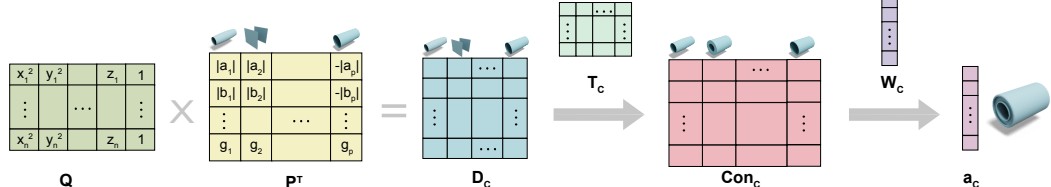

Figure 4: Visualization example shows how the notations (**D**, **Con**, **a**) of the cover branch are obtained.

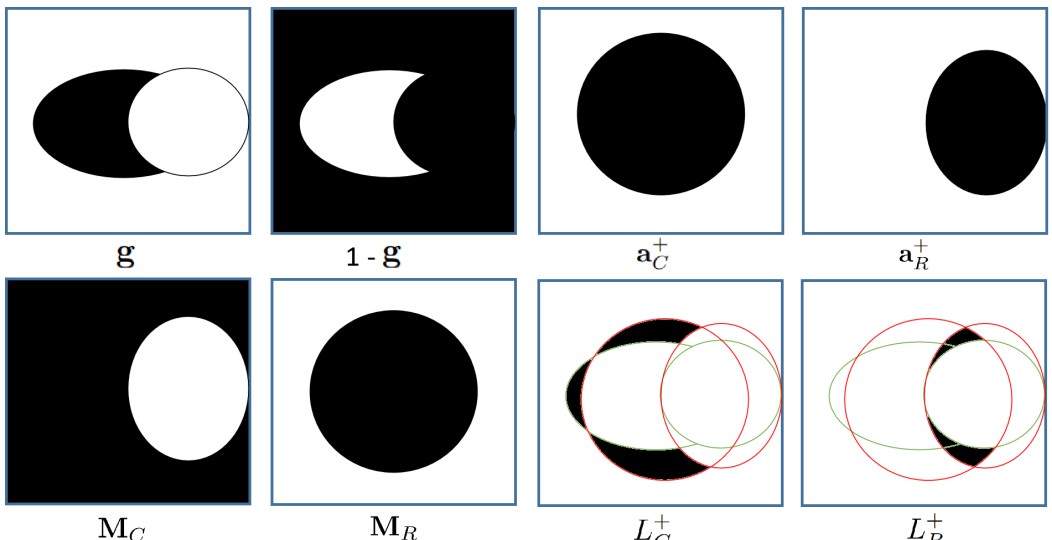

Figure 5: 2D examples for notations used in the loss functions (Equation (6) and (7) in the main paper). Each sub-figure represents a 2D implicit filed defined by the notation below. The inside areas are colored as black and outside areas are colored as white. As for $L_C^+$ and $L_R^+$, the ground truth shape contour is colored as green and the reconstructed shape contour at training is colored as red. The loss functions are utilized to minimize the black area in $L_C^+$ and $L_R^+$.

In Fig. 4, we show how the notations (**D**, **Con**, **a**) of the cover branch are obtained. Notations in the residual branch can be calculated in the similar way. We also provide the 2D examples of notations in the Equation (6) and (7) in the Fig. 5, we can see that $L_C^+$ and $L_R^+$ can force the cover shape to cover the ground-truth shape and make the residual shape to carve out residual volumes.

## C   CAD Mesh Construction and Shape Editing

During inference, after obtaining $\mathbf{P}_C$, $\mathbf{P}_R$, $\mathbf{T}_C$ and $\mathbf{T}_R$, we combine the two sets of primitives that the network has produced to create cover shape $S_C$ and residual shape $S_R$ along with the learned CSG operations in two CSG branches, then perform difference operation between them to output a CAD mesh. Specifically, we obtain the mesh for each primitive by performing Marching-Cube on the signed distance field produced by the quadric equation of that primitive. The CSG operations utilized to assemble the primitive meshes are from the PyMesh library.

To facilitate shape editing in popular CAD software, once we have acquired the primitives and CSG assembly operations, we further convert quadric primitives into basic primitives (e.g., cubes, spheres, cylinder, etc.) by identifying the best-fitting basic primitive for each predicted quadric primitive. This basic primitive fitting algorithm is adopted from SPFN [2]. Subsequently, we can export the primitive parameters and CSG sequence into an OpenSCAD script. Fig. 6 shows more shape editing results. We also provided some our generated OpenSCAD code examples in the supplementary material.

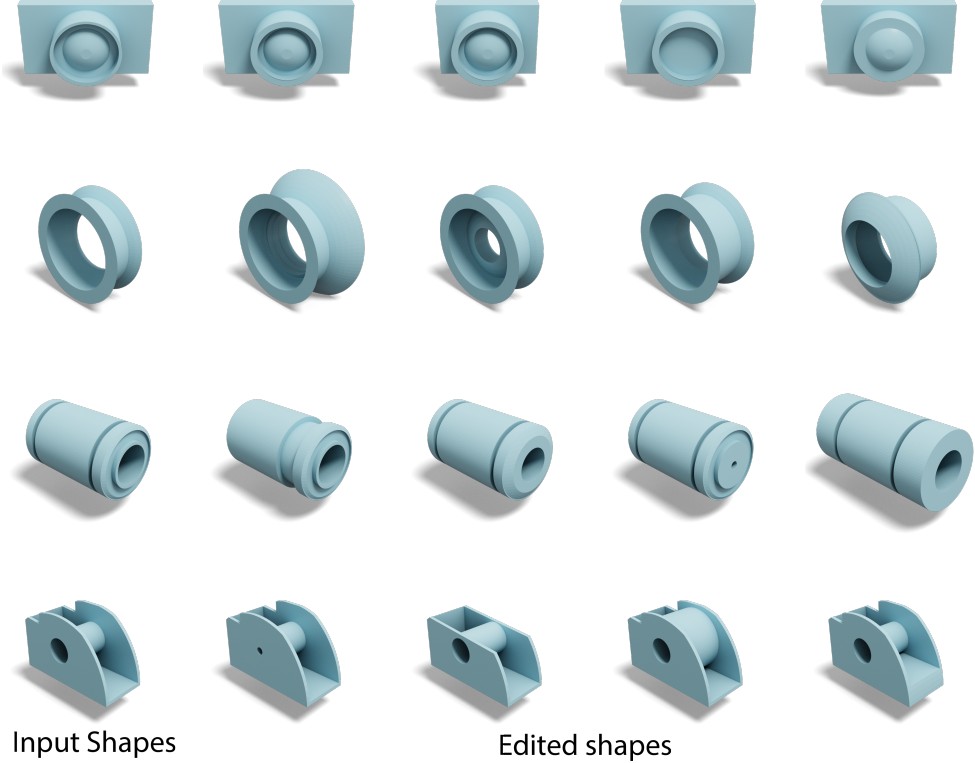

Input Shapes                    Edited shapes

Figure 6: Shape editing results for ABC shapes. D$^2$CSG learns OpenSCAD scripts for a given shape and supports editability.

## D    Additional Results

**Additional metric.**    We have used two additional compactness metrics in the Table 3 of the paper: the number of intermediate shapes and the surface segments. The surface segments is achieved by a hierarchical surface decomposition following the learned CSG tree and it can also measure the compactness of the learned CSG tree. Firstly, we sample 2,048 points on the ground truth shape surface and segment the points according to the point to left and right shape distances. Then we decompose each point cloud segment from the last step according to the point to intermediate shape distances. Finally, we further decompose the point cloud segments from the last step according to the point to primitive distances. The final number of point cloud segments is calculated as the number of surface segments we used in the paper. Based on the fact that $\#CSG = \#intersection + \#union + \#differ = (\#P - \#IS) + (\#IS - 2) + 1 = \#P - 1$ for D$^2$CSG, the number of CSG operations is only related to the number of primitives, and the surface segments is also strongly related to the number of primitives, we mainly use the number of primitives as compactness metric in other tables.

**Pre-training.**    To demonstrate the pre-training on the train set is not helpful for D$^2$CSG, we adopt the same 3D convolutional encoder from CAPRI-Net as 3D shape encoder. We produce the shape feature vector from the input 3D shape voxels in resolution $64^3$ instead of sampling the shape feature vector from the Gaussian distribution. We pre-train the encoder and decoder the in the 5,000 shapes train set from the ABC dataset provided in the CAPRI-Net.

The results are shown in Table 1, where we can see that CAPRI-Net still under-performs compared to D$^2$CSG w/o pre-training. The results also demonstrate that the pre-training step is not helpful to D$^2$CSG since there are many structure and topology variations in the train set and the network cannot learn a good shape prior. In addition, D$^2$CSG can quickly fit the test shape from scratch owing to its more general representation capabilities. Another observation is that the models with pre-training tends to use less primitives than models w/o pre-training, we believe the reason is that pre-trained

Table 1: Ablation study and comparison to CAPRI-Net in the same setting with or without pre-training.

| | Pre-training | | w/o Pre-training | |
|---|---|---|---|---|
| Methods | CAPRI | $D^2$CSG | CAPRI | $D^2$CSG |
| CD $\downarrow$ | 0.177 | 0.198 | 0.183 | **0.069** |
| NC $\uparrow$ | 0.903 | 0.904 | 0.907 | **0.928** |
| ECD $\downarrow$ | 3.990 | 4.170 | 3.918 | **3.091** |
| #P $\downarrow$ | 66.26 | **26.62** | 77.15 | 28.62 |

Table 2: Additional ablation study on the key components of $D^2$CSG: complementary primitives (CP), dual branches (DB), and dropout (DO).

| Row ID | CP | DB | DO | CD $\downarrow$ | NC $\uparrow$ | ECD $\downarrow$ | #P $\downarrow$ | #IS $\downarrow$ |
|---|---|---|---|---|---|---|---|---|
| 1 | - | - | - | 0.183 | 0.907 | 3.92 | 77 | 9.2 |
| 2 | - | ✓ | - | 0.114 | 0.918 | 2.97 | 37 | 10.5 |
| 3 | - | ✓ | ✓ | 0.127 | 0.914 | 3.56 | 32 | 10.0 |

CSG layers have many parameters close to zero which limits the primitive count used in the following optimization stages.

**Additional ablation studies.** We show additional ablation studies in the Table 2. The results again prove that the dual branch design is helpful to achieve more accurate reconstruction results. In addition, the dropout design can produce a more compact CSG sequence while sacrificing the reconstruction accuracy slightly.

**The primitive choice.** We further demonstrate the superiority of quadric primitives over basic primitives in our framework, such as spheres, cylinders, cubes and cones. We adopt the same primitive representations as CSG-Stump to replace the quadric primitives in $D^2$CSG while keeping the other designs unchanged. Since the basic primitives make the network training much slowe (25 minutes of each stage), we only train it in the first stage and compare it to results of $D^2$CSG at the first stage. The results are shown in Table 3, we can see that the quadric primitives can produce better results than basic primitives. We believe there are two reasons: 1. the basic primitives will introduce many complex and non-differentiable operations in point-to-primitive distance calculation; 2. the quadric primitive representation can cover many basic primitives, therefore the change between different primitives is smooth during training, which makes the network use more accurate primitives.

**Effects of the sampled points density on stage 0.** In the mesh to CSG experiments, we sampled different number of training points to examine the effects. Table 4 shows that our results can be slightly improved with more sampled points. This also proves that the number of points does not affect results much when the sampled points number is larger than twenty thousand.

**The hyper parameters.** We examine the effects of several hyper parameters: the dropout threshold $\sigma$, the number of maximum primitives $p$ and intermediate shapes $c$, and the random sample seeds.

The results for dropout threshold $\sigma$ are shown in Table 5, we can see that changing this parameter will slightly affect the results. Specifically, increasing the threshold will produce more compact results but sacrifice the accuracy.

Since the actual primitives and intermediate shapes used by each shape is much less than our pre-set maximum primitive count $p$ and intermediate shape count $c$, we try to reduce $p$ and $c$ and examine

Table 3: Effects of the primitive choice.

| Methods | CD $\downarrow$ | NC $\uparrow$ | ECD $\downarrow$ |
|---|---|---|---|
| Basic | 0.112 | 0.922 | 15.361 |
| Quadric | **0.063** | **0.946** | **4.265** |

Table 4: Effects of the sampled training points density on stage 0.

| Density | CD ↓ | NC ↑ | ECD ↓ |
|---|---|---|---|
| 14,336 | 0.065 | 0.932 | 4.929 |
| 28,672 | 0.063 | 0.946 | 4.265 |
| 57,344 | **0.062** | **0.954** | **4.182** |

Table 5: Effects of the dropout threshold $\sigma$.

| $\sigma$ | CD ↓ | NC ↑ | ECD ↓ | #P | # IS |
|---|---|---|---|---|---|
| 1 | **0.0686** | **0.9344** | **2.986** | 31.12 | 6.11 |
| 3 | 0.0692 | 0.9283 | 3.091 | 28.62 | 5.68 |
| 5 | 0.0713 | 0.9278 | 3.164 | **28.48** | **5.59** |

the effects. As it is showed in Table 6, reducing $p$ and $c$ will make the network use less primitives but result in worse reconstruction quality and more intermediate shapes. In addition, we found reducing $p$ especially affects the reconstruction quality of the shapes with complex topologies, which requires more primitives to reconstruct the details.

We also provide additional Mesh-to-CSG representation learning results on ABC dataset, with respect to different random seeds used for shape feature and network initialization, seed values used in seed #1, seed #2 and seed #3 are set to be 0, 100 and 10,000, respectively. Please see Table 7, our results will only be slightly changed under different random seed settings.

**Additional visualization results for ablation studies.** We show additional visualization results for ablation studies in Fig. 7. The results show that incorporating complementary primitives enables better generalization to shapes with complex structures. The dual branch design further enhances reconstruction accuracy. The dropout design can improve the compactness of the learned sequence while slightly affect the reconstructed visual results.

**Complete CSG Tree.** We show additional complete CSG trees obtained by our method in Fig. 8. The learned CSG tree looks visually compact and natural for the given input shapes.

**Additional visual comparison Results.** We also present more qualitative comparison results on CSG representation learning from meshes and point clouds on ABC dataset and ShapeNet in Fig. 9, 10, 11, 12, additional CSG trees comparison in Fig. 13, 14, 15, 16.

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

Table 6: Effects of the maximum primitive count and intermediate shapes count.

| Row | $p$ | $c$ | CD ↓ | NC ↑ | ECD ↓ | #P ↓ | #IS ↓ |
|---|---|---|---|---|---|---|---|
| 1 | 32 | 8 | 0.167 | 0.914 | 4.315 | **23** | 6.1 |
| 2 | 128 | 16 | 0.089 | 0.926 | 3.371 | 28 | 6.0 |
| 3 | 512 | 32 | **0.069** | **0.928** | **3.091** | 29 | **5.7** |

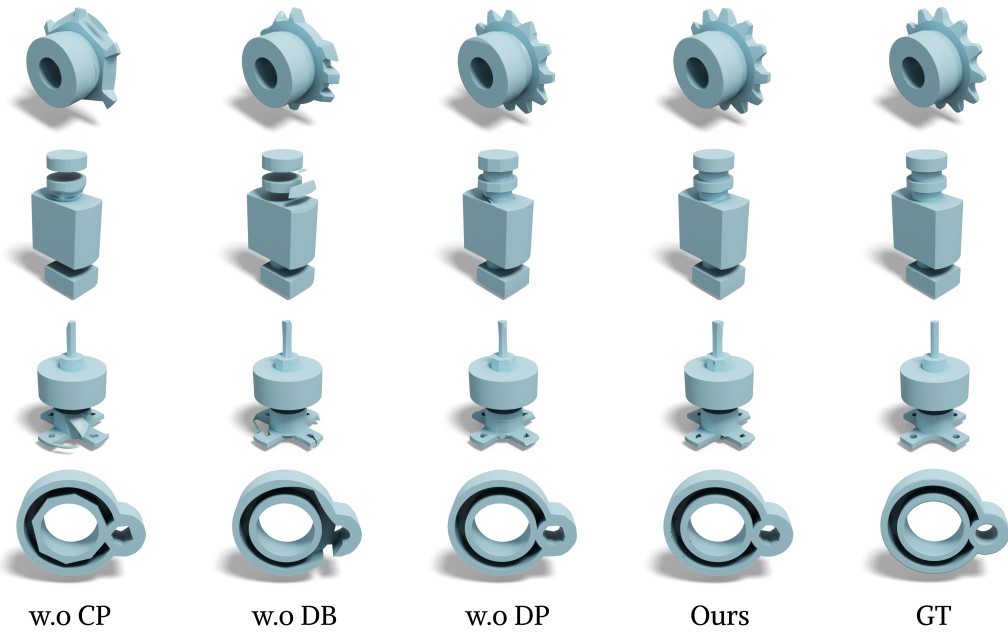

w.o CP       w.o DB       w.o DP       Ours       GT

Figure 7: Visualization results for the ablation study.

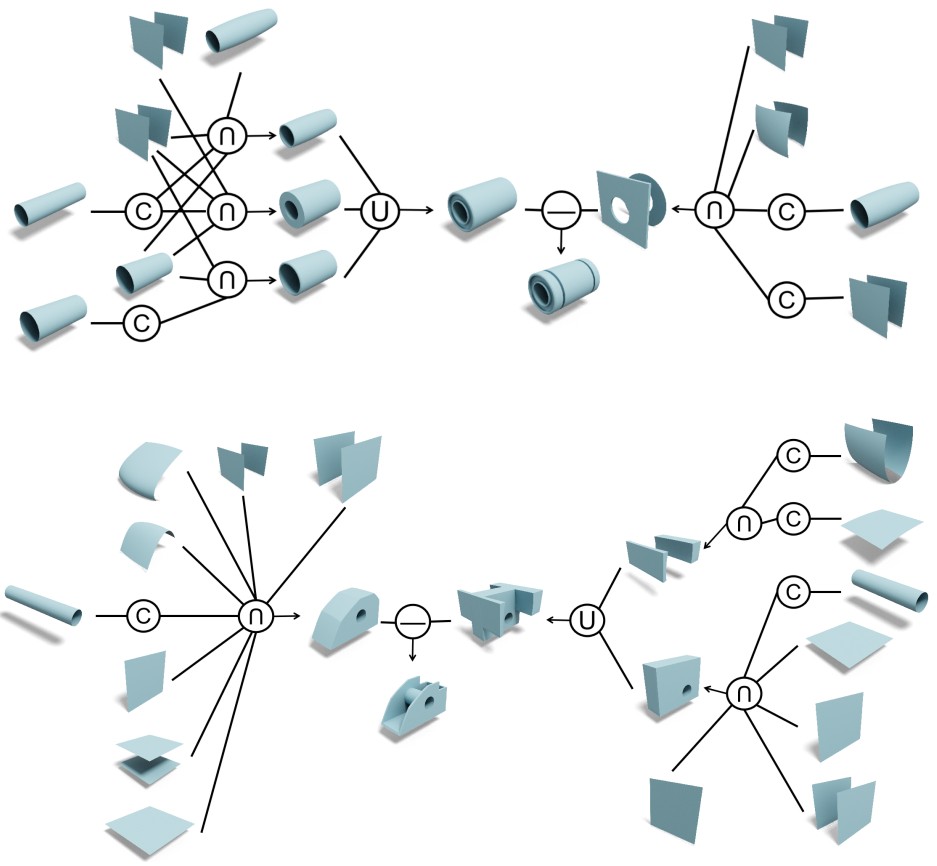

Figure 8: Complete CSG trees obtained by our method.

Table 7: Effects of the random seeds.

| Seeds | CD ↓ | NC ↑ | ECD ↓ | #P ↓ | #IS ↓ |
|---|---|---|---|---|---|
| seed # 1 | 0.069 | 0.928 | 3.091 | 29 | 5.7 |
| seed # 2 | 0.071 | 0.928 | 3.124 | 33 | 6.0 |
| seed # 3 | 0.067 | 0.929 | 3.078 | 32 | 5.8 |

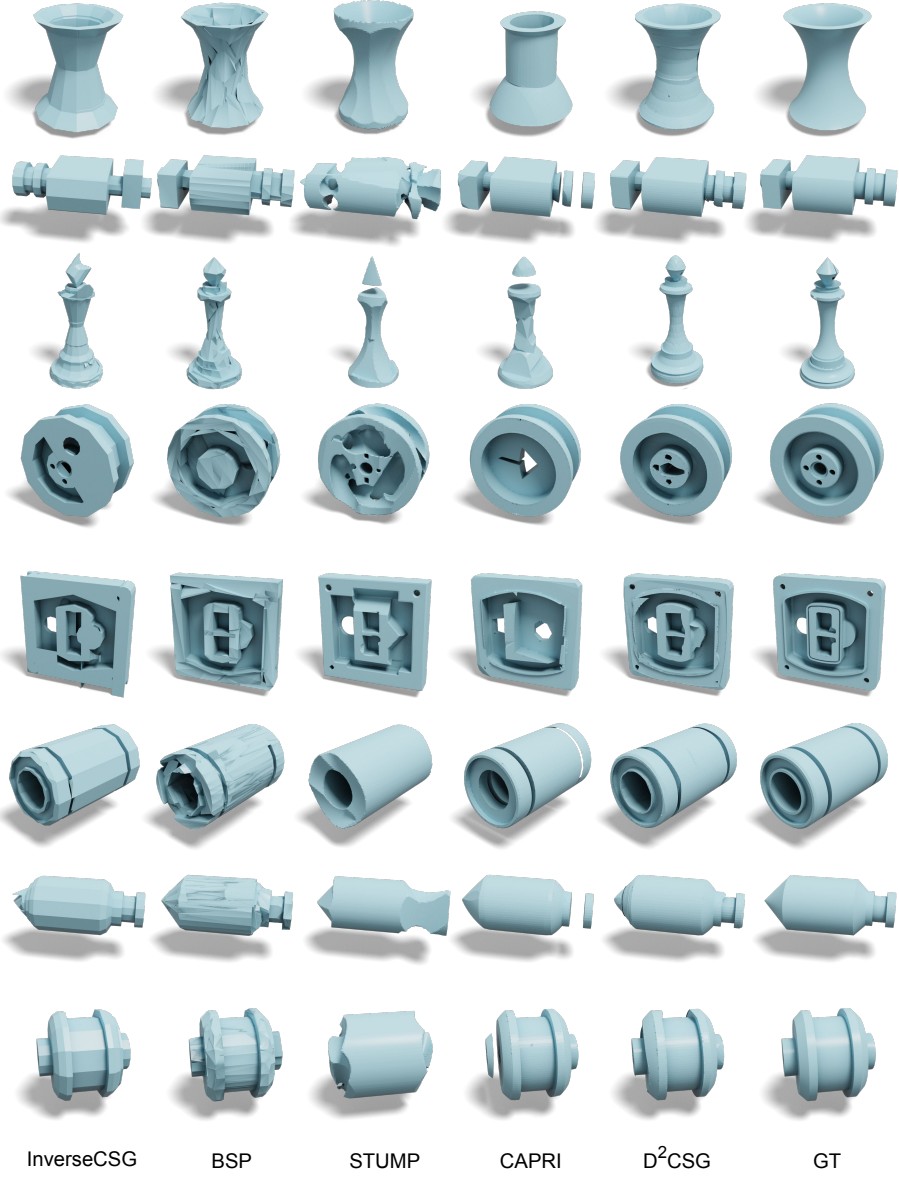

InverseCSG     BSP     STUMP     CAPRI     D$^2$CSG     GT

Figure 9: Comparing CSG representation learning from 3D meshes on ABC.

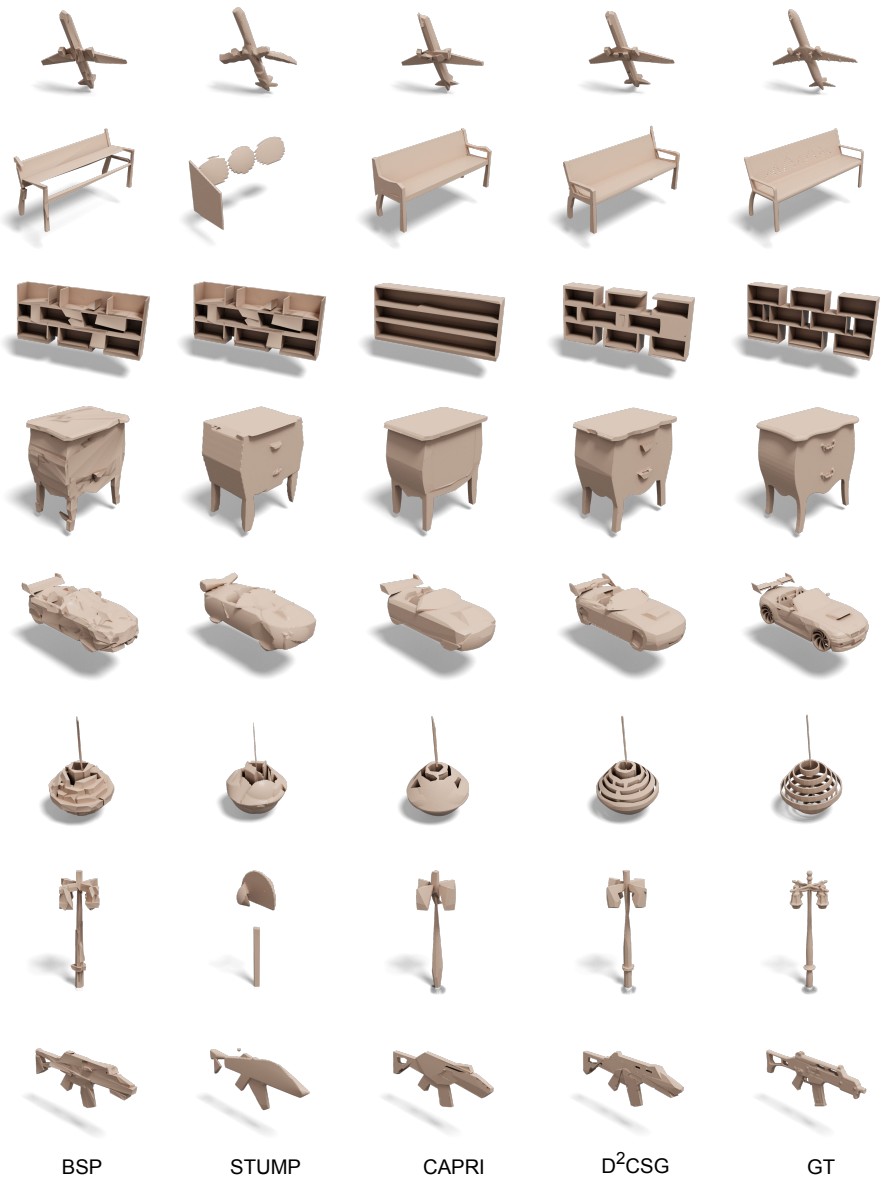

Figure 10: Comparing CSG representation learning from 3D meshes on ShapeNet.

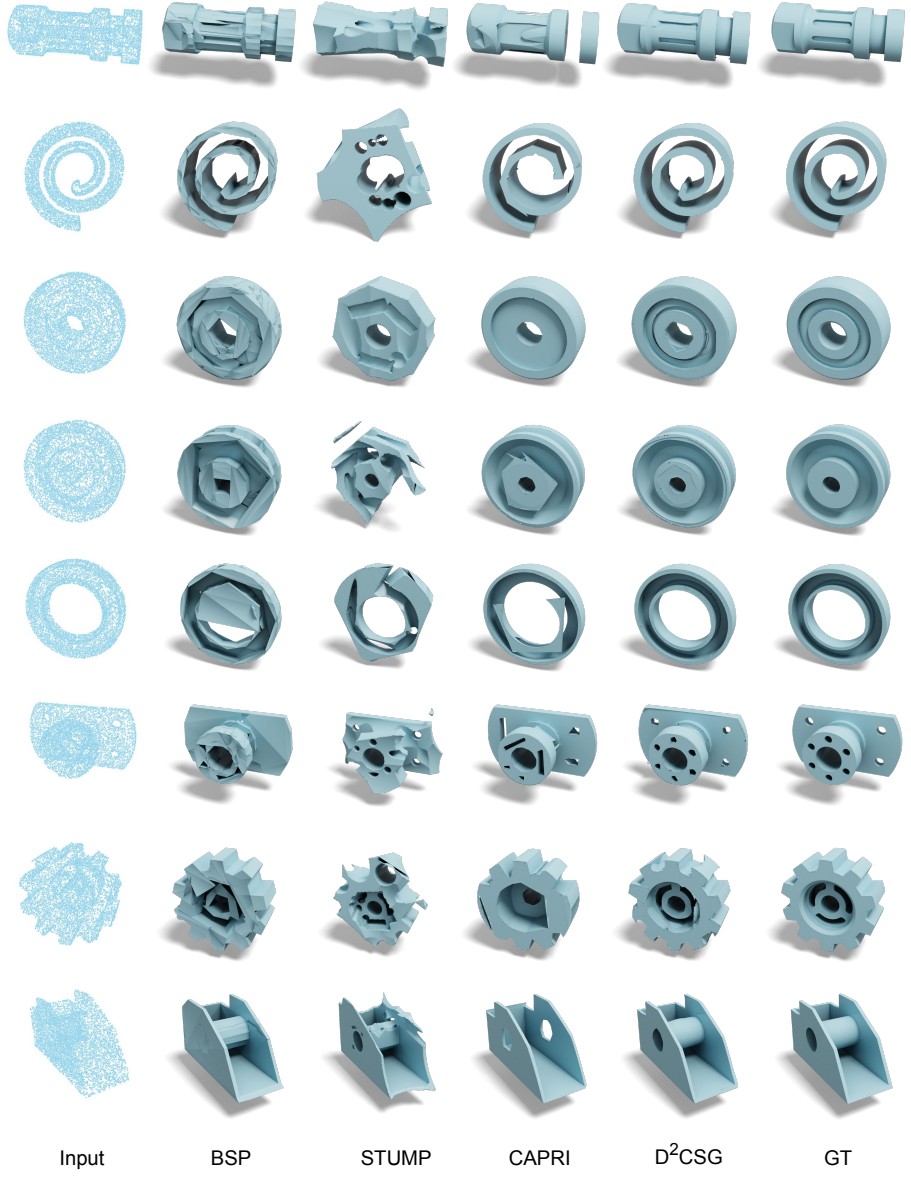

Figure 11: Comparing CSG representation learning from 3D PointCloud in ABC.

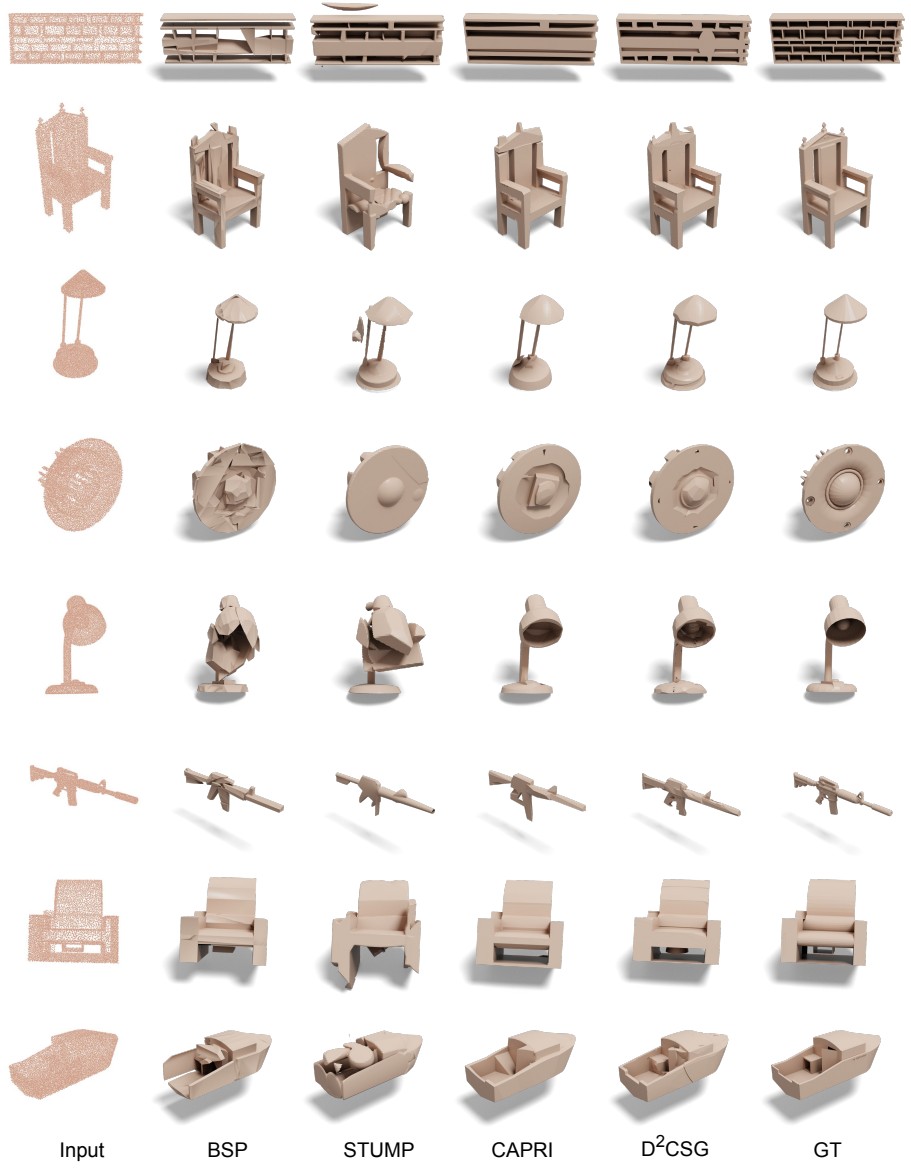

Input BSP STUMP CAPRI D$^2$CSG GT

Figure 12: Comparing CSG representation learning from 3D PointCloud in ShapeNet.

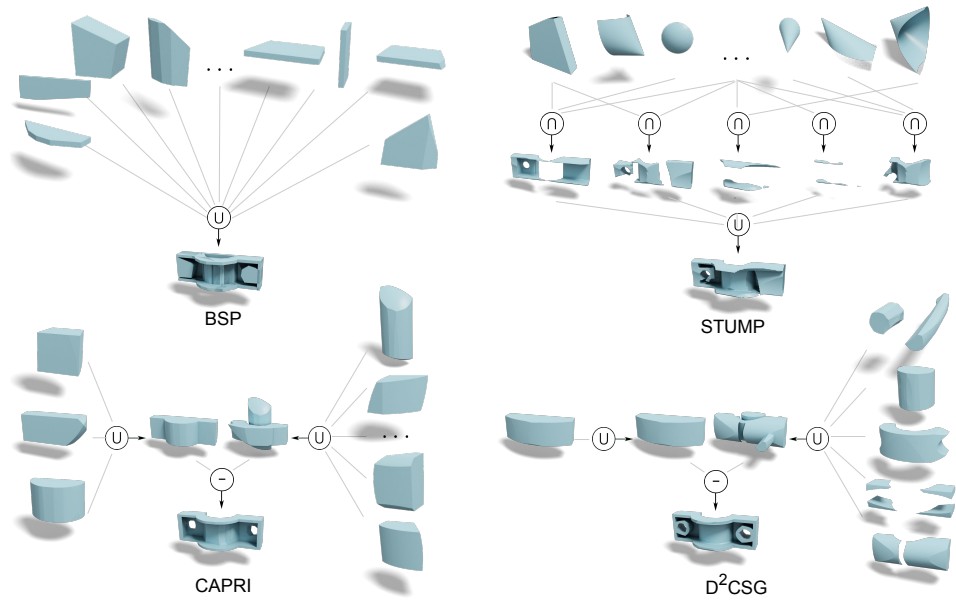

Figure 13: Comparing learned CSG Trees from an 3D example in ABC.

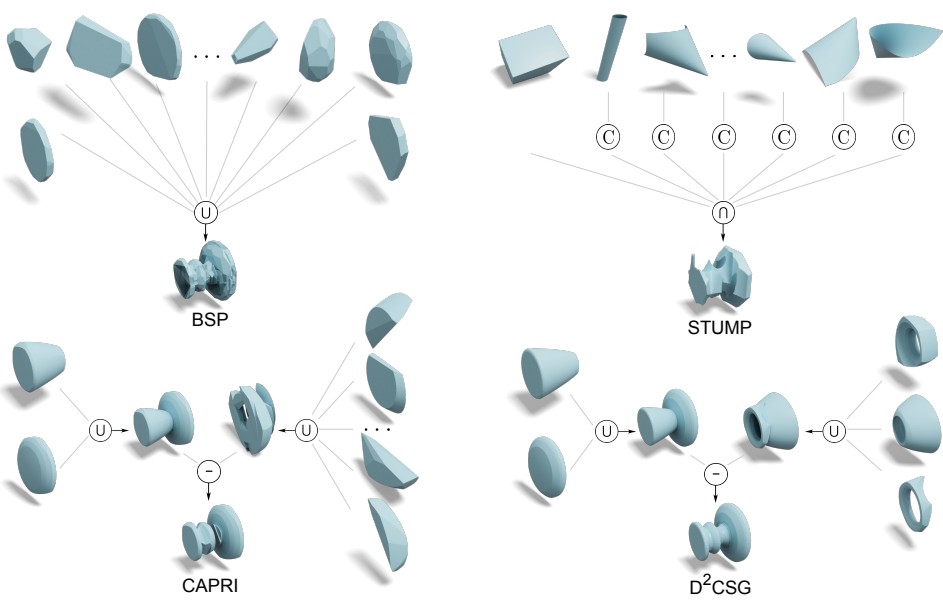

Figure 14: Comparing learned CSG Trees from an 3D example in ABC.

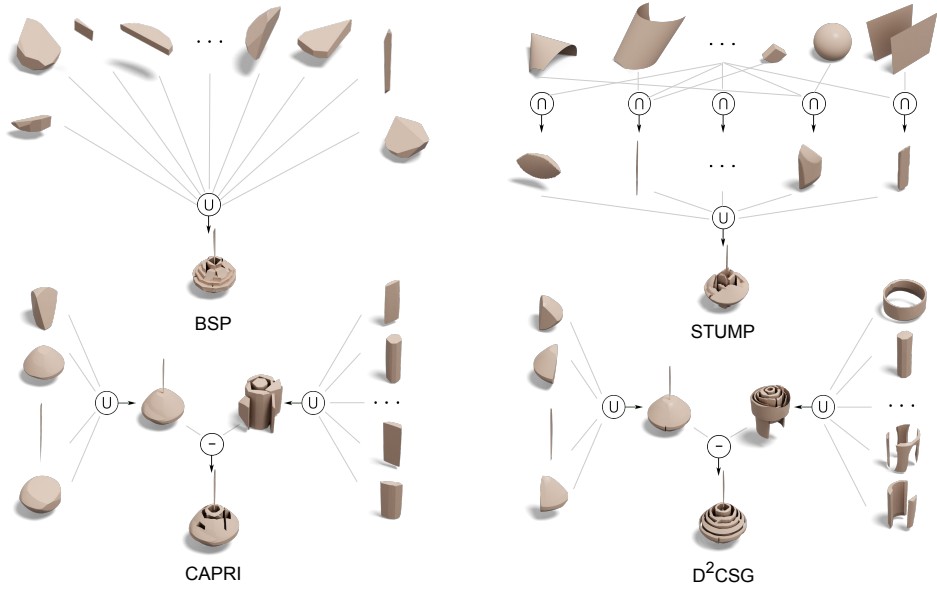

Figure 15: Comparing learned CSG Trees from an 3D example in ShapeNet.

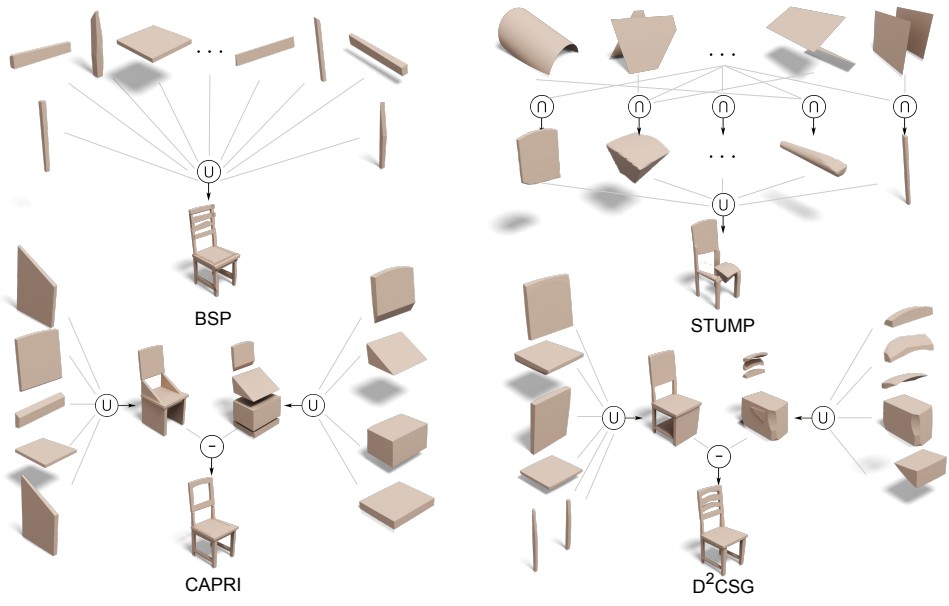

Figure 16: Comparing learned CSG Trees from an 3D example in ShapeNet.