# OpenReview forum: "D$^2$CSG: Unsupervised Learning of Compact CSG Trees with Dual Complements and Dropouts"
_NeurIPS.cc/2023/Conference — NeurIPS 2023 poster_

### Official Review · Reviewer_T4uV · 2023-06-30

**Soundness:** 4 excellent
**Presentation:** 3 good
**Contribution:** 3 good
**Rating:** 7
**Confidence:** 5

**Summary:**

D2CSG presents a neural architecture for inferring CSG programs that reconstruct complex 3D shapes. The architecture is composed of two branches, a cover branch and a residual branch, which are differenced from one another to form the complete shape. The work is largely an extension of, and a strong improvement over, CAPRI-Net, wherein the major differences are: (i) the use primitive complements, (ii) separate primitive sets for the cover and residual branches, (iii) a third training stage, termed dropout, encouraging program sparsity, and (iv) switching from an encoder to an auto-decoder framework. Finding a CSG program to represent a target shape involves "overfitting" the network in a test-time optimization scheme, with an occupancy-based reconstruction loss (with regularizing terms, and various relaxations). Compared to previous neural relaxation approaches for CSG program inference, D2CSG net find programs that result in much better reconstructions (both quantitatively and qualitatively).

**Strengths:**

The paper presents compelling, and well-supported, evidence that D2CSG provides a marked improvement for the task of 3D CSG program inference (and more generally shape abstraction). The insights that contributed to this improvement, while not a dramatic departure from past work, are sound and sensible.

Based on the qualitative results, and quantitative metrics, its clear that D2CSG outperforms a reasonable set of baseline methods, offering a new state-of-the-art bar for CSG program inference. The ablation experiments, in both the main paper and supplemental, are very helpful and generally well-structured. The fact that D2CSG is able to capture a provably larger set of possible shapes than CAPRINet, due to the use of complements in the primitive set, is also a nice property.

**Weaknesses:**

Probably the biggest weakness of the paper is that its methodological improvements are largely incremental on top of the CARPINet system. In my opinion, this shouldn't disqualify the paper from being accepted. Despite the similarity to CARPINet, D2CSG offers real, supported improvement, and would be useful, and of interest, to the community. This type of paper requires strong experimental results, in the form of comparisons to related work and well-crafted ablation conditions, both of which are provided. To give a higher rating, beyond "accept", it would be useful to show that the insights found useful in improving CAPRINet to D2CSG could be extended to other architectures / systems (e.g. they would have more general applicability than this one system). While I think this could be possible, its by no means a given.

```Ablation issues```

There were a couple of small questions/concerns I had surrounding the ablation experiments:

- Is the top row of table 3 just CAPRINet (from the tables this looks like the case). But CAPRINet also has the encoder/decoder vs auto-encoder difference, so I would expect this to be a different architecture (e.g. the CARPINet without pretraining, provided in the supplemental). Please clarify what this row is supposed to report.

- I think it would be important to add rows to Table 3, where just DB, and just DO, are added to row 1. It would be fine to report these results in the supplemental material, but having access to their performance would help disentangle the effects of each change between systems.

```Clarity```

While I wouldn't say that the paper was unclear, I think the overall clarity of the method could be improved. The paper is not very self-contained, much of the technical details are omitted from the main paper, and pushed to either past-work or the supplemental material. Due to space constraints, a full explanation of the training scheme and architecture is likely impossible to have in the main paper, but I think the main paper could still be improved to give a better *motivation* for why the architecture and training scheme is designed in the way that it is.

- Most helpful would be to add more details and annotations to Figure 2 (e.g. the D, Q, T, Con, and a matrices). If these can't fit in the figure, then sub-figures showing parts of the pipeline should be introduced (e.g. to the supplemental).

- In the text, introductory paragraphs should be used to set the stage for the various matrices, and what information they are suppsoed to hold. For instance, there is no explanation given for what the "Con" matrix (first introduced on line 175) holds, beyond the formulas for how it is constructed. An additional paragraph at the start of 3.2, walking through all of the matrices that will be created (and what information they hold) would be very helpful.

- Similarly, while some technical details (though not all) are provided in 3.3 indicating how the loss functions change through the training stages, it would benefit the paper to have a high-level section at the beginning of 3.3 explaining *why* multiple stages are needed (while this reasoning might have been provided in past-work, it needs to be restated here, for self-containment). For instance, something like: in stage 0 no hard decisions are made for either (a) primitive to intersection members or (b) which convexes are added into union, then in stage 1 we make a hard decision for (b), and in stage 2 we make a hard decision for (a).

```Minor```

- It would be useful to know how the reconstruction metrics change when converting from quadrics to basic primitives for user-editing (especially compared with just directly optimizing basic primitives, and not quadrics).

- Some discussion of related work on shape program inference can be improved. [36] is not a supervised method, and should likely be grouped with [11], as RL-based methods for unsupervised shape program inference (see [1*]). Relatedly, [2*] should also be added to this section, as an alternative to per-shape optimization for 3D CSG inference.

- typos/phrasing:
  -  L:27, genera
  - L:163, are float


[1*] Neurosymbolic Models for Computer Graphics, Eurographics 2023 STAR

[2*] PLAD: Learning to Infer Shape Programs with Pseudo-Labels and Approximate Distributions, CVPR 2022

**Questions:**


- If I understood the dropout operation, its based on finding changes to the primitive / intersection contributions that make little difference (e.g. a small delta) to the output binarized shape from the network. I'm surprised that it isn't instead based in deltas to the networks reconstruction loss (e.g. difference versus the target shape). Did you explore any types of dropout operations that considered the target occupancy values?

- The conclusion has the following statement:  ```We also have ample visual evidence that the CSG trees obtained by our method tend to be more natural than those produced by prior approaches```. Looking through the decompositions in the supplemental, I'm unconvinced that many people would posit the decompositions as `natural' -- I would consider either adding additional evidence to support this claim, or removing it.

- The conclusion implies that for datasets like ABC, that lack a consistent global part decomposition, 'overfitting' is more acceptable, with the subtext that its not helpful to learn over more than one shape at a time. I would encourage the addition of a little more nuance to this statement: while ABC-type shape are certainly less globally consistent compared with other domains (like ShapeNet chairs), mechanical parts share clear commonalities with one another, especially at the sub-component/local level, and other methods have found success in learning over distributions of these types of shapes (e.g. [46]). Thus, instead of tying the the merits of `overfitting' to this line of argument, it would be better to let it stand on its own.


- (minor) I'm about confused about Eq 7. From the explanation of the method, I would have thought that W is ignored after stage 0 is done. In table 1, it looks like it is not used at all in stage 1, but then re-introduced (and binarized) in stage 2. Line 109 in the supplemental indicates that stage 2 training is the same as stage 1, but how can this be the case if W is re-introduced (is it frozen in stage 1, then unfrozen/modified for stage 2)? I would request additional clarification on this mechanism.




**Limitations:**

While the `overfitting' setup certainly seems to improve reconstruction performance (at least for D2CSG, though interestingly not for CAPRINet), there are certain downsides of moving away from a shared latent space: e.g., ill-conditioned inference tasks (e.g. program from partial point cloud, or program from image), and shape to shape interpolation, are no longer possible. It would be good for the paper to discuss the impact of this design decision a bit (beyond the change in reconstruction).

---

> ### Author Rebuttal · Authors · 2023-08-08
>
> **it would be useful to show that the insights found useful in improving CAPRINet to D2CSG could be extended to other architectures / systems.**
>
> **A:** The dual complementary idea can be applied to other primitive-based methods for better concavity reconstruction. For example, ExtrudeNet only uses the union of extrusions to reconstruct a shape; it is possible to introduce complement extrusions to better handle concavity. The dropout idea can also be applied to other assembly-based systems as long as they employ intersection or union assembly operations.
>
> * ExtrudeNet: Unsupervised Inverse Sketch-and-Extrude for Shape Parsing, ECCV 2022
>
> **Ablation issues.**
>
> **A:**
> * Good point, we show CAPRI-Net with pre-training and encoder at the first row here, while other rows are performed without pre-training. It would be better to show CAPRI-Net without an encoder here, see updated Table 2 in the PDF file attached to the global response
> * The existing rows can prove the efficiency of our designs. We also add two additional rows and visualization results to help disentangle the effects of each change, see results in the PDF file attached to the global response.
>
> **Clarity.**
>
> **A:**
> * We can revise Figure 2 as suggested.
> * We can add introductory paragraphs for the various matrices.
> * We can provide high-level details in Section 3.3 about the multi-stage training.
>
> **It would be useful to know how the reconstruction metrics change when converting from quadrics to basic primitives for user-editing.**
>
> **A:** We will provide them in the supplementary material.
>
> **Some discussion of related work on shape program inference can be improved.**
>
> **A:** We will improve them as suggested.
>
> **Q1: Did you explore any types of dropout operations that considered the target occupancy values?**
>
> **A:** The dropout module is designed to remove primitives/intermediate shapes that have little effect on the reconstructed shape. The $\Delta S$ is calculated based on the difference between the reconstructed shapes before and after dropout. We did not calculate $\Delta S$ between the target shape and the reconstructed shape since it is hard to determine a proper threshold. If we set it smaller, then many complicated shapes would not be considered for dropouts since they usually have a large difference to the target shape. if we set the threshold larger, then many primitives modeling shape details could be dropped as the number of query points is less around such details. In our design, all shapes will have the chance to be considered for dropouts while keeping the shape after dropouts close to what it was before.
>
> **Q2:  I'm unconvinced that many people would posit the decompositions as `natural'**
>
> **A:** We can tone this down in the revision.
>
> **Q3:  I would encourage the addition of a little more nuance to this statement.**
>
> **A:** Thanks for your advice. We will reword the conclusion and add the nuances as the reviewer described by mentioning that "some of the models in ABC may not possess sufficient generalizability in their primitive assemblies" to avoid giving the false impression that all ABC models including mechanical objects lack commonalities in structure.
>
> **Q4:  I'm confused about W in Eq 7.**
>
> **A:** At stage 0, values in W are set close to 1 (line 221). At stage 1, W is not used and is not updated. At stage 2, W is re-introduced and updated (line 223). Although the reconstruction loss $L_{rec}^*$ is the same in both stages, the $a^*$ in $L_{rec}^*$ is derived from different equations for stage 1(Eq 3) and stage 2 (Eq 7). In addition, the gradients from reconstruction loss $L_{rec}^*$ won’t be used to update W at stage 2, W is only updated by dropout (line 223) at stage 2.
>
> **Q5:  Discuss more impact about using overfitting.**
>
> **A:** We will add such discussions as suggested. Thank you.

---

> > ### Comment · Reviewer_T4uV · 2023-08-14
> >
> > I thank the authors for their detailed and well-written response. I remain very positive on this paper, and would like to see its inclusion to the conference.

---

### Official Review · Reviewer_AcKp · 2023-07-05

**Soundness:** 3 good
**Presentation:** 3 good
**Contribution:** 3 good
**Rating:** 6
**Confidence:** 4

**Summary:**

The paper presents an unsupervised network learning method for reconstructing CSG trees from CAD models. The network is an enhancement of CAPRI-NET, and features the fixed operations (from bottom to top) of intersection -> union -> difference on primitives modeled by quadratic surfaces, where the difference is always applied on two intermediate shapes produced from subtrees. The enhancement to CAPRI-NET is the allowance of difference in the form of inverse primitives before intersection, which the paper shows to cover all possible CSG trees in contrast to the incomplete coverage of CAPRI-NET. In addition, the paper uses dropout pruning to reduce the redundant primitives or intersected intermediate shapes. Through experiments on ABC and ShapeNet datasets and comparisons with previous works on unsupervised CSG reconstruction, this paper shows improved results with good reconstruction accuracy and compactness. Ablation studies further show the usefulness of the dual branch design, inverse primitives, and dropout pruning.

**Strengths:**

The paper shows the limitation of a previous work CAPRI-NET in representing all possible CSG trees and fixes it with a simple addition of inverse primitives.

The paper introduces dropout pruning to improve compactness of result CSG trees.

Extensive tests have shown the new algorithm can recover more compact and faithful results than previous works.

**Weaknesses:**

It's not very clear how the dropout pruning is applied in the training process. Is it used in the last step and after its application there will be no network finetuning? Or the interleaved application of dropout and finetuning can be more helpful? It's desirable that the authors provide a detailed study on this issue.



**Questions:**

In addition to questions above, there are some detailed questions:

1. Could the primitives before intersection be shown in the expanded CSG trees? That would help readers better understand the complexity and details of the results.

2. Is there any intuitive understanding of the learned weighting vector W in Eq(4)?

3. Why $\Delta S$ of Eq(6) is not normalized by the number of sample points $n$?

4. Line223, should it refer to Eq(3) instead?



**Limitations:**

The authors have discussed limitations in generalization, detail recovery and limited expressiveness of quadratic primitives.

---

> ### Author Rebuttal · Authors · 2023-08-08
>
> **It is not very clear how the dropout pruning is applied in the training process.**
>
> **A:** Dropouts are not applied in the last step but during the training process; please refer to lines 221-222 in the paper. After dropouts, the network parameters will be tuned. We iteratively perform this process until the maximum number of iterations has been reached or when no primitive/intermediate shapes are dropped.
>
> **Q1: Could the primitives before intersection be shown in the expanded CSG trees.**
>
> **A:** Since the complicated shapes in our supplementary material usually use many primitives, we only show a part of the obtained CSG trees. We show the primitives before intersection for the simple shape in our pipeline, as shown in Figure 2 of the main paper. We also show additional primitive visualization examples, see Figure 1 in PDF file attached to the global response.
>
> **Q2: Is there any intuitive understanding of the learned weighting vector W in Eq(4)?.**
>
> **A:** Values in W are learned close to 1 after stage 0. This setup will help the union layer avoid using the min operation in Eq(3)  at stage 0 and all intermediate shapes could have gradients in the early stage.
>
> **Q3: Why $\Delta S$ of Eq(6) is not normalized by the number of sample points.**
>
> **A:** Since the number of sampled points is constant, normalization will not change the learning process. In our experiments, we sampled a fixed number query points from the shape before dropouts at stage 2 to calculate $\Delta S$. We will clarify this part in the revision.
>
> **Q4: Line223, should it refer to Eq(3) instead?.**
>
> **A:** Yes, Line 223 is equation (3). Thanks for spotting this mistake.

---

> > ### Comment · Reviewer_AcKp · 2023-08-15
> >
> > Thanks for the responses.

---

### Official Review · Reviewer_DMnr · 2023-07-07

**Soundness:** 4 excellent
**Presentation:** 3 good
**Contribution:** 3 good
**Rating:** 6
**Confidence:** 5

**Summary:**

The paper proposes a reconstruction approach to building constructive solid geometry (CSG) from other 3D modalities like meshes and point clouds. The key contribution of the paper is a dual representation that considers both the shape and its complement that are built with Boolean intersection and union operations with a set of primitive convex quadric surfaces and their inverses. Two branches in a fully differentiable neural network are responsible for generating the shape and the complement that is Boolean subtracted to output the final CSG representation of a target shape. The dual representation is general and enables complex shapes to be captured well as demonstrated by extensive results.

**Strengths:**

The paper is well written and easy to follow. It offers a competitive solution to an important inverse problem in CAD, and the proposed representation is novel, general and backed by strong empirical results in two standard datasets. The design choices are well justified and evaluated. The decision to overfit to a target shape rather than attempting to generalize is sensible, but see also a related point in weaknesses.

**Weaknesses:**

One weakness of choosing an overfitting approach as opposed to learning from datasets is challenges related to robustness to noise and outliers. The method appears to be sensitive to how clean the input geometry is, and with noisy pointclouds or low resolution meshes, it is very likely for unintended sliver geometry to be constructed. This is apparent in Figure 4, although the proposed method does perform better than others. Some discussion on this would be beneficial in the limitations or future work section.

Another weakness is the choice of using quadric surfaces as primitives. CAD shapes are typically built with prismatic primitives like planes, cylinders, etc. and while quadric surfaces can represent most of such surfaces, optimizing in this representation is unlikely to reconstruct the exact prismatic primitives as seen in the results throughout the paper.

**Questions:**

In the related work under Deep CAD Models, it appears that BRepNet, UV-Net and SB-GCN are attributed as reverse engineering models while these are all encoders.

Some missing citations that might be worth adding:
- https://dl.acm.org/doi/abs/10.1145/3528223.3530078
- https://dl.acm.org/doi/abs/10.1145/3550469.3555424

References [46] and [47] are duplicates


**Limitations:**

Yes

---

> ### Author Rebuttal · Authors · 2023-08-08
>
> **One weakness of choosing an overfitting approach as opposed to learning from datasets is challenges related to robustness to noise and outliers.**
>
> **A:**  Yes, unintended geometry could be produced for noisy point clouds or low-resolution meshes. This is a general issue for overfitting-based methods. Incorporating D$^2$CSG with other point cloud denoising or upsampling modules would consititute interesting future work for robust CSG reconstruction. We can add this discussion in the revision.
>
> **Another weakness is the choice of using quadric surfaces as primitives.**
>
> **A:**  We did discuss the limitation of quadric primitives in the conclusion. CAD shapes are not only constructed by prismatic primitives but also by other complex primitives, such as NURBS. However, we consider D$^2$CSG as a key step toward learning general and compact CSG assembly sequences. The quadric primitives in D$^2$CSG can be easily changed to other primitives, and we find quadric primitives can achieve the best fitting results compared to prismatic primitives; see Table 2 in the supplementary material. In addition, when the shape can be approximated with simple primitives, we can convert/approximate quadric functions with simple primitives (please refer to line 306).
>
> **Q1: Revise related works and add citations.**
>
> **A:** Thank you for these suggestions. We will incorporate them in the revision.

---

> > ### Comment · Reviewer_DMnr · 2023-08-15
> >
> > Thanks for the response. I am retaining my score and would be happy to see this paper accepted.

---

### Official Review · Reviewer_dmBa · 2023-07-07

**Soundness:** 4 excellent
**Presentation:** 2 fair
**Contribution:** 3 good
**Rating:** 6
**Confidence:** 2

**Summary:**

This paper proposed a method for unsupervised learning of CSG trees from mesh or point cloud. An auto-decoder approach is used, i.e., each training shape is represented by a learned latent code. Compared to previous approach CAPRI-Net, the proposed approach used complementary primitives and dual branch design to represent shapes in more complicated and accurate CSG primitives. Also, the proposed approach used dropout during training to avoid redundancy of the inferred CSG tree. The method is validated on ABC dataset and ShapeNet.

**Strengths:**

- The idea of complementary primitives and dual branch are novel and effective.
- The proposed method significantly outperforms previous method both qualitatively and quantitatively.
- The experiment section is conducted extensively, which supports the claims well.

**Weaknesses:**

- Some important technical details are not clear (see questions).
- It will be easier to follow the paper if there are some visualizations explaining the insights of complementary primitives and dual branch.

**Questions:**

- How is the T matrix predicted from the network?
- The residual loss appears in the figure-2 of main paper, it will be helpful to move its definition to the main paper.
- Many components of the paper is based on CAPRI-Net, it will be helpful to provide a compact background section for CAPRI-Net in the paper.

---

> ### Author Rebuttal · Authors · 2023-08-08
>
> **It will be easier to follow the paper if there are some visualizations explaining the insights of complementary primitives and dual branch.**
>
> **A:** We showed additional visualization results for the ablation study, see Figure 2 in the PDF file attached to the global response.
>
> **Q1: How is the T matrix predicted from the network.**
>
> **A:** The T matrix is not predicted from the network but set as learnable parameters in the network.
>
> **Q2: The residual loss appears in the figure-2 of main paper, it will be helpful to move its definition to the main paper..**
>
> **A:** Thank you for pointing out this issue. We put details of the cover loss and residual loss in the supplementary material. We will mention how we use the cover loss and residual loss in the main paper.
>
> **Q3: Many components of the paper is based on CAPRI-Net, it will be helpful to provide a compact background section for CAPRI-Net in the paper.**
>
> **A:** We plan to add this compact background section to briefly introduce the components adopted from CAPRI-Net as space allowed.

---

### Official Review · Reviewer_rfjZ · 2023-07-10

**Soundness:** 4 excellent
**Presentation:** 4 excellent
**Contribution:** 4 excellent
**Rating:** 8
**Confidence:** 3

**Summary:**

The authors propose a novel, unsupervised method to reconstruct the CSG tree given a 3D shape. The authors prove that all CSG trees can be formulated with a boolean difference operation as the last step. Therefore, to generate the final reconstructed shape, the proposed method uses two branches, cover and residual, to produce the shapes used to keep and remove in the boolean difference operation. To further enhance the model's capability of generating all possible shapes, the inverse convex primitives are introduced to incorporate the boolean difference operations other than the last step. The proposed method also improves compactness using dropout. The results demonstrate significant improvement over prior works qualitatively and quantitatively.

**Strengths:**

1. The proposed method significantly improves the reconstruction quality, which can be easily observed from the qualitative results in Fig 3.

2. The paper is well-written and provides most of the required details (both in the main text and supplementary).

3. The authors demonstrate that the model can reconstruct the CSG trees from general and complex 3D shapes in both ABC and ShapNet datasets, and provide shape editing capability for downstream CAD tasks.

**Weaknesses:**

1. The generated CSG trees might not look natural to designers and engineers since they usually have some shared patterns and preferences when modeling 3D shapes. Moreover, the modeling sequences are often related to the design and manufacturing intents. Therefore, it would be great to see user study results comparing the generated CSG trees with real modeling sequences (e.g., Fusion 360 Gallery Reconstruction Dataset).

2. Designers usually do not model 3D shapes simply using primitives. Most solid modeling tools in CAD are profile-based, meaning that users draw 2D profiles and use them to perform 3D operations such as extrude, revolve, sweep, chamfer, fillet, etc. Therefore, mapping the generated CSG trees into those operations will be necessary for practical use.

3. Point cloud representation can be challenging when reconstructing large objects with small details, such as the camping car example in Fig. 3.

**Questions:**

1. Stages 0, 1, and 2 in the main text are not coherent with stages 1, 2, and 3 in Table 1.

2. How does the proposed method perform on multiple objects (or objects with disconnected parts)?

3. How do the point sampling density, quality, or methods affect the results?

4. Have the authors tried generating CSG trees with depths larger than two (e.g., using an iterative approach)? Is there any way to justify whether the current approach is sufficient to generate any shapes? In other words, do shapes generated in each branch have enough complexity?

**Limitations:**

Yes, the limitations are addressed.

---

> ### Author Rebuttal · Authors · 2023-08-08
>
> **The generated CSG trees might not look natural to designers.**
>
> **A:** We generally agree with the reviewer and there is definitely *more* that is left to do to get there.
>
> One intrinsic difficulty is that there is no (single) ground truth for the CSG assembly of a given CAD shape. Each 3D shape could be represented by different modeling sequences, and different artists might have different design preferences: some may like the additive style, while others more used to the subtractive style. In addition, different modeling sequences also have different editing complexity for different modeling targets. Taking a more objective view, we currently focus on accounting for and striving towards the reconstruction accuracy and compactness of the CSG assembly when designing our network.
>
> **User study results comparing the generated CSG trees with real modeling sequences (e.g., Fusion 360 Gallery Reconstruction Dataset).**
>
> **A:** This is an interesting thought. However, the extrusion-based modeling sequences in Fusion360 are quite different from our primitive-based CSG assembly. The latter is our goal. We acknowledge that sketch-and-extrude is quite a natural editing and modeling paradigm for designers, e.g., see Google (Trimble) SketchUP. Learning those types of assemblies would involve a very difficult goal, e.g., see ExtrudeNet and SECAD-Net as recent attempts.
>
> * ExtrudeNet: Unsupervised Inverse Sketch-and-Extrude for Shape Parsing, ECCV 2022
>
> * SECAD-Net: Self-Supervised CAD Reconstruction by Learning Sketch-Extrude Operations, CVPR 2023
>
> **Designers usually do not model 3D shapes simply using primitives.**
>
> **A:** A fair point; see our remarks above on sketch-and-extrude. To this end, we believe that D$^2$CSG could serve as a valuable supplementary approach to existing modeling paradigms. By utilizing extrusions and sweep operations to create primitives, we can employ D$^2$CSG to construct even more intricate shapes. Introducing broader and more versatile primitives into the D$^2$CSG framework presents an intriguing avenue for future research. We intend to include this discussion in our paper.
>
> **Point cloud representation can be challenging.**
>
> **A:**  Yes, point clouds are not the best representations for details. One future work is to first use SOTA point cloud upsampling methods to produce details or use adaptive sampling to sample more points around details.
>
> **Q1: Stages 0, 1, and 2 in the main text are not coherent with stages 1, 2, and 3 in Table 1.**
>
> **A:** Thank you for pointing out this issue. We will address it in the revision.
>
> **Q2: How does the proposed method perform on multiple objects (or objects with disconnected parts)?**
>
> **A:** Our method is self-supervised and is robust to unseen shape structures and categories; see Figure 3. In particular, we use a shape-specific optimization technique, allowing the method to effectively accommodate diverse objects, including multi-part objects.
>
> **Q3: How do the point sampling density, quality, or methods affect the results?**
>
> **A:** Our method is no exception to the "garbage in, garbage out" concept. Less sampled points or low-quality points will produce worse results, see Table 1 in the PDF file attached to the global response.
>
> **Q4: Have the authors tried generating CSG trees with depths larger than two (e.g., using an iterative approach)? Is there any way to justify whether the current approach is sufficient to generate any shapes? In other words, do shapes generated in each branch have enough complexity?**
>
> **A:** In the supplementary material, we provide a proof that the current approach is able to produce any shape. Please see Proposition 2 and its proof in the supplementary material (line 35).

---

> > ### Comment · Reviewer_rfjZ · 2023-08-21
> >
> > Thank the authors for the rebuttal. I would love to see the primitive-based method and extrude-based method converge in the future. I will remain my rating as a strong accept.

---

### Author Rebuttal · Authors · 2023-08-08

We thank all the reviewers for their insightful comments and encouraging remarks. We are glad to see reviewer recognitions that our approach is “novel,” “effective,” "compelling,” and “significantly” outperforms existing methods. Since the reviewer questions were all technical in nature, seeking more details and clarifications, we will answer them in the individual responses. The submitted PDF file attached to the global response includes additional results.

Code and data will be released upon paper acceptance.

---

### Decision · Program_Chairs · 2023-09-21

**Decision:**

Accept (poster)

**Comment:**

Overall, the reviewers appreciated the novelty of the presented approach and the significant improvements over prior methods. The reviewers had several remarks on how the paper should be modified:
- a brief background section on CAPRI-Net
- revised motivation for choice of architecture and training scheme
- revised explanation of various matrices and rationale for various stages
- revised related work section

Given that these changes will be incorporated as promised in the rebuttal, the reviewers and the AC have no objection to accepting the paper.